# Structural changes in the shallow and transition branch of the Brewer-Dobson circulation induced by El Niño

Mohamadou Diallo[1,2], Paul Konopka[1], Michelle L. Santee[3], Rolf Müller[1], Mengchu Tao[1], Kaley A. Walker[4], Bernard Legras[2], Martin Riese[1], Manfred Ern[1], and Felix Ploeger[1,5]

[1]Institute of Energy and Climate Research, Stratosphere (IEK–7), Forschungszentrum Jülich, 52 425 Jülich, Germany.
[2]Laboratoire de Météorologie Dynamique, UMR8539, IPSL, UPMC/ENS/CNRS/Ecole Polytechnique, Paris, France.
[3]Jet Propulsion Laboratory, California Institute of Technology, Pasadena, California, USA.
[4]Department of Physics, University of Toronto, Toronto, Ontario, Canada.
[5]Institute for Atmospheric and Environmental Research, University of Wuppertal, Wuppertal, Germany.

**Correspondence:** Mohamadou Diallo (m.diallo@fz-juelich.de)

**Abstract.**

The stratospheric Brewer-Dobson circulation (BD-circulation) determines the transport and atmospheric lifetime of key radiatively active trace gases and further impacts surface climate through downward coupling. Here, we quantify the variability in the lower stratospheric BD-circulation induced by the El Niño Southern Oscillation (ENSO), using satellite trace gas measurements and simulations with the Lagrangian chemistry transport model, CLaMS, driven by ERA-Interim and JRA-55 reanalyses. We show that despite discrepancies in the deseasonalised ozone ($O_3$) mixing ratios between CLaMS simulations and satellite observations, the patterns of changes in the lower stratospheric $O_3$ anomalies induced by ENSO agree remarkably well over the 2005–2016 period. Particularly during the most recent El Niño in 2015–2016, both satellite observations and CLaMS simulations show the largest negative tropical $O_3$ anomaly in the record. Regression analysis of different metrics of the BD-circulation strength, including mean age of air, vertical velocity, residual circulation and age spectrum, shows clear evidence for structural changes of the BD-circulation in the lower stratosphere induced by El Niño, consistent with observed $O_3$ anomalies. These structural changes during El Niño include a weakening of the transition branch of the BD-circulation between about 370–420 K ($\sim$100–70 hPa) and equatorward of about 60° and a strengthening of the shallow branch at the same latitudes and between about 420–500 K ($\sim$70–30 hPa). The slowdown of the transition branch is due to an upward shift in the dissipation height of the large-scale and gravity waves, while the strengthening of the shallow branch results mainly from enhanced gravity wave breaking in the tropics-subtropics combined with enhanced planetary wave breaking at high latitudes. The strengthening of the shallow branch induces negative tropical $O_3$ anomalies due to enhanced tropical upwelling, while the weakening of the transition branch combined with enhanced downwelling due to the strengthening shallow branch leads to positive $O_3$ anomalies in the extratropical upper troposphere-lower stratosphere (UTLS). Our results suggest that a shift of the ENSO basic state toward more frequent El Niño-like conditions in a warmer future climate will substantially alter UTLS trace gas distributions due to these changes in the vertical structure of the stratospheric circulation.

## 1 Introduction

The lower stratosphere (10–25 km) is a key region in a changing climate. In this region, the amount of key greenhouse gases, such as water vapor and ozone, which radiatively impact temperatures both locally and globally, are regulated by advection, mixing, and chemistry (e.g. Forster and Shine, 2002, 1999; Solomon et al., 2010; Riese et al., 2012; Dessler et al., 2013). Ozone is a greenhouse gas, which is mainly produced in the stratosphere (10–50 km), and is directly regulated by the upwelling strength of the stratospheric circulation in the tropics.

The stratospheric mean meridional circulation, the so-called Brewer-Dobson circulation (e.g. BD-circulation; Brewer, 1949; Butchart, 2014), is defined as a slow circulation in which air parcels rising in the tropics drift poleward in the stratosphere and are transported downward at high latitudes via its shallow and deep branches (Bönisch et al., 2011; Lin and Fu, 2013). Driven by wave breaking in the stratosphere (Haynes et al., 1991; Rosenlof and Holton, 1993; Newman and Nash, 2000; Plumb, 2002) and varying on subseasonal to decadal timescales, the BD-circulation is modulated by natural variability (Plumb and Bell, 1982; Trepte and Hitchman, 1992; Niwano et al., 2003; Punge et al., 2009), including the El Niño Southern Oscillation (ENSO) (Randel et al., 2009).

ENSO is a coupled atmosphere-ocean phenomenon occurring in the equatorial Pacific Ocean with drastic changes in regional sea surface temperatures (SSTs), impacting surface weather and climate (e.g. Bjerknes, 1969; Cagnazzo and Manzini, 2009; Wang et al., 2016). ENSO alternates between anomalously warm (El Niño) and cold (La Niña) conditions in the tropical Eastern or Central Pacific Ocean at intervals of $2-8$ years (Philander, 1990; Baldwin and O'Sullivan, 1995). El Niño and La Niña events are associated with variations in tropical SSTs, convection, and atmospheric temperature as well as in the circulation throughout the global troposphere (L'Heureux et al., 2017; Scaife et al., 2017). During El Niño, the Eastern equatorial or Central Pacific Ocean is anomalously warm and convection is shifted towards this region (e.g., Avery et al., 2017). During La Niña, in contrast, highest SSTs and most intense convection occur in the western Pacific. In either phase, the fluctuations associated with ENSO usually last for a little longer than one year. The oscillations in SSTs of the Pacific Ocean are accompanied by displacements of tropospheric temperature and precipitation patterns around the globe (Brönnimann et al., 2007).

ENSO is also a major mode of climate variability that affects the variability of the BD-circulation. Most of the previous research on ENSO influences on the stratosphere has concentrated on tropical and extratropical temperatures as well as on planetary waves in the extratropics and on polar vortex stability during El Niño based on global circulation models and observations (e.g., Sassi et al., 2004; Manzini et al., 2006; Taguchi and Hartmann, 2006; Garcia-Herrera et al., 2006; Garfinkel and Hartmann, 2007; Calvo et al., 2008; Ineson and Scaife, 2009; Butler et al., 2014). A substantial part of the interannual variability in the lower stratosphere turns out to be related to ENSO (Randel et al., 2009; Calvo et al., 2010). El Niño events directly warm the troposphere and cool the tropical lower stratosphere with a node near the tropopause, suggesting a tropical coupling of the tropospheric and stratospheric variability (Calvo-Fernandez et al., 2004; Randel et al., 2009; Mitchell et al., 2015). Analyses of atmospheric temperatures from satellite observations indicated an overall warming of the tropical tropo-

sphere superimposed on equatorially symmetric subtropical Rossby wave gyres during El Niño events (Yulaeva and Wallace, 1994; Calvo-Fernandez et al., 2004). Using a comprehensive high-top general circulation model to investigate the dynamical mechanisms involved during ENSO winters, Simpson et al. (2011) concluded that the response in tropical upwelling is predominantly driven by anomalous transient synoptic-scale wave drag in the southern hemisphere subtropical lower stratosphere.

Based on zonally averaged satellite observations, Randel et al. (2009) found negative ozone and temperature anomalies in the tropical lower stratosphere attributed to strengthening tropical upwelling of the BD-circulation during El Niño events. In contrast, La Niña events induce an opposite zonal mean effect (e.g. Calvo et al., 2010; Konopka et al., 2016). Climate models show that the ENSO modulations of the tropical upwelling appear to be linked to different propagation and dissipation patterns of parameterized gravity waves during winter (Garfinkel and Hartmann, 2008; Calvo et al., 2010; Simpson et al., 2011). Accord-

ing to Konopka et al. (2016), the variability of tropical upwelling in the lower stratosphere shows strong regional variations in the zonally resolved picture, especially during strong La Niña years when planetary wave activity at levels directly above the tropical tropopause is enhanced and the subtropical jets are significantly disturbed.

Most previous studies of direct ENSO influence on the BD-circulation have focused on changes in the strength of the tropical upwelling and on the mechanisms (wave-mean flow interaction) that produce its acceleration or deceleration (Randel

et al., 2009; Calvo et al., 2010; Konopka et al., 2016). Here, we investigate the detailed changes in the vertical structure of different BD-circulation branches based on satellite observations and simulations with the Chemical Lagrangian Model of the Stratosphere (CLaMS) (McKenna et al., 2002; Konopka et al., 2004; Pommrich et al., 2014). Birner and Bönisch (2011) found a separation in the residual circulation transit times (RCTT) between the shallow and deep branches of the BD-circulation. In particular, they found much smaller transit times into the mid-latitude than into the polar lowermost stratosphere. Based on

these findings, the shallow branch is found in the tropical stratosphere and in the lower mid-latitudinal stratosphere equatorward of about 60° below 500 K ($\sim$ 30 hPa), whereas the deep branch is found throughout the high-latitude stratosphere poleward of 70° and above 500 K. In addition, Lin and Fu (2013) further separated the shallow branch defined by Birner and Bönisch (2011) into two sub-branches: the transition branch (i.e. between 370–420 K (100–70 hPa)) and the shallow branch (i.e. between 420– 500 K (70–30 hPa)). Here, we use this definition of the branches to identify the "fingerprints" of the ENSO-induced variability

in the structure of the BD-circulation. We disentangle the changes in each branch of the BD-circulation related to ENSO using multiple regression analysis on different diagnostic quantities derived from the satellite observations, CLaMS simulations and meteorology of two modern reanalysis products included in the SPARC Reanalysis Intercomparison Project (S-RIP) (Fujiwara et al., 2017). A description of the satellite observations, model data and the multiple regression technique is included in Section 2. Section 3 shows the ENSO impact on simulated and observed ozone mixing ratios in the lower stratosphere. Section 4

presents an analysis of the ENSO-induced changes in the vertical structure of the BD-circulation in the lower stratosphere, based on mean age of air, vertical velocity, residual circulation and age spectrum diagnostics. Finally, we discuss a possible dynamical mechanism for these changes in the vertical structure of the circulation and potential impacts on decadal and long-term changes (Sect. 5).

## 2 Data and Methodology

### 2.1 Description of the CLaMS model

The Chemical Lagrangian Model of the Stratosphere (CLaMS) is a Lagrangian transport model with trace gas transport based on the motion of 3-D forward trajectories and an additional parameterization of subgrid scale atmospheric mixing (McKenna et al., 2002; Konopka et al., 2004). The CLaMS model allows ozone concentrations to be simulated through a simplified formulation of stratospheric chemistry (Pommrich et al., 2014). The lower boundary values for ozone mixing ratio are set to zero in the lowest model layer (roughly the boundary layer), while the upper boundary condition ($\sim$500 K) is imposed based on mean climatological satellite fields. For this study, we carried out simulations with the CLaMS model driven by 6-hourly horizontal winds and diabatic heating rates both from ERA-Interim (ERA-I) (Dee et al., 2011) and Japanese 55-year Reanalysis (JRA-55) (Kobayashi et al., 2015) reanalyses, respectively provided by the European Centre for Medium-Range Weather Forecasts and the Japan Meteorological Agency. For the wind and temperature fields, CLaMS uses $1° \times 1°$ for the horizontal resolution and the native reanalysis vertical resolution. The mean vertical resolution of air parcels in the CLaMS Lagrangian model is about 400 m near the tropopause. The simulation driven by ERA-I covers the 1979–2016 period, whereas the simulation driven by JRA-55 covers the 1979–2013 period. Both reanalyses are described in detail by Fujiwara et al. (2017) for the S-RIP project, which is a coordinated inter-comparison of modern global atmospheric reanalyses.

### 2.2 Lower stratospheric $O_3$ from CLaMS and Aura-MLS

To analyze the response of the BD-circulation to ENSO variability, we use ozone ($O_3$) mixing ratios and different diagnostics of the stratospheric circulation strength, as described in the following. The simulated $O_3$ mixing ratios from the CLaMS set-up used in this work were previously analysed by Pommrich et al. (2014) for validating the CLaMS simulations. In addition, the $O_3$ mixing ratios from CLaMS simulations driven by ERA-I and JRA-55 are sampled at the MLS measurement geolocations to avoid sampling bias during the inter-comparisons. Reliable agreement with satellite observations has been found regarding seasonality as well as variability related to the Quasi-Biennial Oscillation (QBO). The first part of the present analysis is a further validation of CLaMS' ability to reproduce interannual stratospheric variability related to ENSO.

The observational data used for comparison with CLaMS simulations are monthly mean $O_3$ mixing ratios in the lower stratosphere from the Aura Microwave Limb Sounder (MLS), covering the period 2005–2016 (Livesey et al., 2017). The MLS instrument, flying aboard the EOS-Aura satellite, is designed to measure a wide range of physical and chemical quantities, including $O_3$ (Waters et al., 2006). The version 4.2abundances MLS data were produced with improved retrieval algorithms, which substantially reduced the occurrence of unrealistically small $O_3$ values at 215 hPa in the tropics observed in the previous version 2.2 MLS product (Livesey et al., 2008). Note that the version 4.2 MLS $O_3$ data used here are not significantly different from the previous version MLS observations at pressures less than 100 hPa, but show less oscillatory behavior and fewer retrieval artifacts induced by cloud contamination in the tropical upper troposphere and lower stratosphere. The version 4.2 $O_3$ data are characterized by a vertical resolution of 2.5$-$3.5 km, a precision of $\pm 0.02-0.04$ ppmv, a systematic uncertainty of $\pm 0.02$–0.05 ppmv $+ \pm 5$–10 % and a lowest recommended level of 261 hPa for individual profile measurements with

a horizontal resolution in the UTLS of $\sim 300$–400 km along the orbital-track line of sight (Livesey et al., 2017; Santee et al., 2017). The regression results will not be affected by these intrinsic uncertainties since they apply to the $O_3$ mixing ratios and not the anomalies. Additional detailed information on the quality of MLS $O_3$ in the upper troposphere-stratosphere in previous versions can be found in dedicated validation papers (Read et al., 2007; Livesey et al., 2008; Froidevaux et al., 2008).

## 2.3 Metrics of the BD-circulation

In addition to the trace gas diagnostics, the strength of the BD-circulation is commonly deduced from age of air related diagnostics, including the mean age of air (AoA) and the age spectrum, and also the residual vertical velocity ($\overline{w^*}$), the residual circulation transit time (RCTT) and the residual circulation mass stream function ($\psi^*$) (Reithmeier et al., 2008; Li et al., 2012; Diallo et al., 2012; Butchart, 2014; Abalos et al., 2015; Ploeger et al., 2015a, b; Ploeger and Birner, 2016). Mean AoA is defined as the average transit time for an air parcel since entering the stratosphere, and is therefore the first moment of the full transit time distribution termed the *age spectrum*. As shown by Hall and Plumb (1994), mean AoA can be calculated in a model from a "clock-tracer" that is an inert tracer with a linear increase in the troposphere or at the surface. Note that we calculate mean AoA and age spectrum relative to the lowest model level following the surface, as this is a common choice in global models (Waugh and Hall, 2002).

The age spectrum includes the detailed transit time information and is advantageous for investigating different transport pathways (e.g., Ploeger and Birner, 2016). In the CLaMS model, the age spectrum is calculated using a total of 60 different boundary pulse tracers, with pulses released in the lowest model layer in the tropics between $15°$ S and $15°$ N, constituting the pulse source region $\Omega$ at source times $t'$. Note that releasing the pulses only in the tropics between $15°$ S and $15°$ N might bias the age spectrum results in the lowermost stratosphere. It is likely that a substantial amount of air originating in the extratropics crosses the tropopause near the subtropical jets, especially during summer and autumn in the northern hemisphere. Since this air is not taken into account, the young portion of the age spectrum is likely being underestimated. For each pulse, the tracer mixing ratio $\chi_i(r,t)$ is set to unity in $\Omega$ for 30 days, and is set to zero in $\Omega$ otherwise. These pulses are released every two months. For instance, the first tracer pulse has its source time in January 1979, the second tracer pulse in March 1979, and so on. The age spectrum is a Green's function or a boundary propagator, $G$, that solves the continuity equation for the mixing ratio of a conserved and passive tracer (Hall and Plumb, 1994). As a function of transit time (elapsed time) $\tau = t - t'_i$, the age spectrum is constructed from these $N$ pulse tracers at each sample field time $t$ and sample region $r$ as $G(r,t|\Omega, t - \tau_i) = \chi_i(r,t)$. For more details about the set-up and calculations see Ploeger and Birner (2016).

The residual circulation transit time (RCTT) is a 2-D diagnostic defined as the transit time of an air parcel through the stratosphere, if it were advected only by the residual circulation, and measures the strength of the residual circulation (Bönisch et al., 2011; Birner and Bönisch, 2011). RCTTs are calculated from 2-D CLaMS backward trajectories driven by the mass-weighted isentropic zonal mean diabatic circulation, and the reference level is set to the 340 K isentrope in the tropics to include transport in the tropical tropopause layer (Fueglistaler et al., 2009a). For more details about the RCTT calculations see Ploeger et al. (2015a). In addition, we analyze the strength of the tropical upwelling related to ENSO using $\overline{w^*}$, calculated from the Transformed Eulerian Mean (TEM) circulation standard formula in geometric coordinates (e.g. equation (3.5), Andrews et al.,

1987) and the diabatic heating rate (Fueglistaler et al., 2009b; Wright and Fueglistaler, 2013) from both reanalyses. In contrast to the integrated residual circulation transit time along the trajectory of an air parcel, $\overline{w^*}$ is a local 2-D quantity.

## 2.4 Multiple regression model

To properly disentangle the ENSO impact on these metrics of the BD-circulation from the other sources of natural variability, the monthly zonal mean $O_3$ mixing ratios and other diagnostic quantities are analyzed by using a multiple regression model as a function of latitude ($\phi$) and altitude (z). This regression method is an established method and appropriate to disentangle the relative influences of the considered climate indices on BD-circulation variability, as it includes time lag coefficients as a function of $\phi$ and z for each proxy, including ENSO signal. For more detail about the method and its further applications see Diallo et al. (2012, 2017, 2018). The regression method decomposes the temporal evolution of a monthly zonal mean parameter, $\chi$, in terms of a long-term linear trend, seasonal cycle, QBO, ENSO, volcanic aerosol and a residual. The model yields for a given parameter, $\chi$ (herein $O_3$, AoA, $\overline{w^*}$, RCTT, $\Psi$, age spectrum, air mass fraction, temperature, zonal mean wind, Eliassen-Palm flux and its divergence)

$$\chi(t,\phi,z) = a(\phi,z) \cdot t + C(t,\phi,z) + \sum_{k=1}^{3} b_k(\phi,z) \cdot P_k(t - \tau_k(\phi,z)) + \varepsilon(t,\phi,z) \tag{1}$$

where $P_k$ represents the predictors or proxies of different atmospheric sources of variability. Thus, $P_1$ is a normalized QBO index (QBOi) from CDAS/Reanalysis zonally averaged winds at 50 hPa, $P_2$ is the normalized Multivariate ENSO Index (MEI) (Wolter and Timlin, 2011) and $P_3$ is the Aerosol Optical Depth (AOD) from satellite data (Vernier et al., 2011; Khaykin et al., 2017; Thomason et al., 2018). The coefficients are a linear trend $a$, the annual cycle $C(t,\phi,z)$, the amplitude $b_1$ and the lag $\tau_1(\phi,z)$ associated with the QBO, the amplitude $b_2$ and the lag $\tau_2(\phi,z)$ associated with ENSO and the amplitude $b_3$ and the lag $\tau_3(\phi,z)$ associated with AOD. The constraint applied to determine the parameters $a$, $b_1$, $b_2$, $b_3$, $\tau_1(\phi,z)$, $\tau_2(\phi,z)$, $\tau_3(\phi,z)$ and $C$ is to minimize the residual $\varepsilon(t,\phi,z)$ in the least squares sense. Because of the presence of lags in the QBO, ENSO and AOD terms in equation (1), the problem is nonlinear and the residual may have multiple minima as a function of the parameters. In order to determine the optimal values of $\tau_1(\phi,z)$, $\tau_2(\phi,z)$ and $\tau_3(\phi,z)$, the residual is first minimized at fixed lag and then sorted out over a range of lags. This is done in sequence for QBO, ENSO and AOD. Here we neglect solar forcing, because our data set covers only one solar period. Uncertainty estimates for the statistical fits are calculated using a Student's t-test technique (Zwiers and von Storch, 1995; Bence, 1995; von Storch and Zwiers, 1999).

## 3 ENSO impact on lower stratospheric $O_3$

Figure 1a shows the interannual variability of the deseasonalized $O_3$ from CLaMS simulations driven by ERA-I and JRA-55 sampled at the MLS measurement geolocations together with MLS observations averaged in the tropical lower stratosphere between 380–425 K as a percentage change relative to the climatological monthly mean mixing ratio during the 2005–2016 period. Generally, a consistent picture of $O_3$ interannual variability emerges between observations and model simulations

driven by ERA-I and JRA-55. Note that the CLaMS $O_3$ values are two time as large as the MLS $O_3$ values and this difference in the magnitude of the $O_3$ anomalies is not due to a sampling bias. The factor of 2 difference in the zonal mean magnitude between CLaMS and MLS $O_3$ anomalies is likely due to the lack of tropospheric $O_3$ chemistry and the $O_3$ lower boundary condition being set to zero in CLaMS, combined with tropical upwelling being too strong and tropical-extratropical exchange being too weak in the model. These different possible reasons for the factor of 2 difference are further discussed in Section 5. The deseasonalized tropical $O_3$ time series exhibit seasonal variations in both model simulations and observations, which are negatively correlated with the Multivariate ENSO Index (MEI) (Fig. 1(a, c)). In particular, during the 2015–2016 period, the deseasonalized $O_3$ shows negative anomalies in the tropical lower stratosphere due to the enhanced tropical upwelling caused by both the extreme El Niño event and the QBO disruption (e.g. easterly wind shear at 100–40 hPa) (Diallo et al., 2018).

However, the overall $O_3$ interannual variability is challenging to interpret because of its regulation by the complex interplay between the ENSO and QBO induced variability (e.g., Taguchi, 2010; Liess and Geller, 2012; Neu et al., 2014; Diallo et al., 2018), by the climate change impact (e.g., Bekki et al., 2013; Iglesias-Suarez et al., 2018; Ball et al., 2018; Wargan et al., 2018) and by the emissions of ozone depletion substances (e.g., Dhomse et al., 2018; Chipperfield et al., 2018; Montzka et al., 2018). Therefore, to elucidate the ENSO impact on the stratospheric $O_3$ anomalies, the multiple regression is performed both without and with explicitly including the ENSO signal.The difference between the residual ($\varepsilon$ in (1)) without and with explicit inclusion of the ENSO signal gives the ENSO-induced impact on stratospheric $O_3$ anomalies. This approach of differencing the residuals is similar to direct calculations, projecting the regression fits onto the ENSO basis functions herein termed the *amplitude variation* ($b_2 \times STD(MEI)$ i.e. term $b_2$ in (1) normalised by the standard deviation of the MEI). For illustration, please see supplementary Fig. 2 and 4 in Diallo et al. (2017) and also Diallo et al. (2018). Figure 1b shows time series of the $O_3$ changes induced by ENSO variability in the tropical lower stratosphere averaged between 380–425 K and estimated from the difference between the residual ($\varepsilon$ in (1)) with and without explicit inclusion of the ENSO signal for the 2005–2016 period. The ENSO-induced variability in lower stratospheric $O_3$ mixing ratios shows a good agreement between CLaMS simulations driven by both reanalyses and MLS observations, though again with a factor of 2 difference in the magnitude. These $O_3$ anomalies show a strong negative correlation with the MEI, reaching $-77.9\%$ for CLaMS driven by ERA-I, $-70\%$ for CLaMS driven by JRA-55 and $-85.7\%$ for MLS.

Figures 2(a-c) show latitude-time series of the ENSO-induced variability in monthly mean $O_3$ mixing ratios in the lower stratosphere and estimated from the difference between the residual ($\varepsilon$ in (1)) with and without explicit inclusion of the ENSO signal for the 2005–2016 period. The patterns of ENSO-induced variability in the CLaMS $O_3$ driven by both reanalyses and MLS observations agree very well, though again with a factor of 2 difference in the magnitude related to the high-biased $O_3$ variability in CLaMS consistent with Fig. 1(a, b). In addition, the gradient in the MLS and JRA-55 $O_3$ anomalies between the tropics and extratropics in the southern hemisphere is smoother than that in CLaMS simulations driven by ERA-I, likely due to its too strong tropical upwelling (Dee et al., 2011; Wright and Fueglistaler, 2013; Abalos et al., 2015). The CLaMS and MLS $O_3$ anomalies are negative in the lower stratosphere during El Niño years (e.g., 2006–2007, 2010–2011, 2015–2016) and positive during La Niña years (e.g. 2008–2009, 2011–2012, 2013–2014), consistent with previous studies (Randel et al., 2009; Calvo et al., 2010; Konopka et al., 2016). In particular, the most recent El Niño event produces an extremely large

negative $O_3$ anomaly in the lower stratosphere, inducing a record anomaly of $-15\%$ in the tropics for MLS (twice as large for CLaMS), consistent with Diallo et al. (2018). This strong increase in the magnitude of negative $O_3$ anomalies is interpreted as a manifestation of the strengthening of the tropical upwelling induced by El Niño (see Sect. 4) (Randel et al., 2009). These substantial $O_3$ anomalies are consistent with recently published strong ozone and water vapor anomalies during the 2015–2016

El Niño (Avery et al., 2017; Diallo et al., 2018). The two consecutive La Niña events in 2011–2012 exhibit the largest positive $O_3$ anomalies in decadal satellite records.

Figures 3(a-c) depict the zonal mean impact of ENSO on $O_3$ variability for CLaMS simulations driven by ERA-I (a) and JRA-55 (b) together with MLS (c) calculated as the projection of the regression fits onto the ENSO basis functions for the 2005–2016 period, i.e. the amplitude variation. There is good agreement between CLaMS and MLS regarding the pattern of $O_3$

variations related to El Niño-like conditions, with the negative $O_3$ anomalies in the JRA-55 simulations much more confined to the tropics. In the tropical UTLS, the negative $O_3$ anomalies during El Niño are due to the enhanced tropical upwelling, transporting upward fresh air poor in $O_3$ from the troposphere. The negative $O_3$ anomalies from simulations driven by ERA-I are stronger than those from MLS and JRA-55, corroborating the too strong upwelling in ERA-I (Dee et al., 2011; **?**). In the extratropical UTLS ($30°$–$70°$), CLaMS simulations driven by both reanalyses together with MLS observations show a related

positive $O_3$ anomaly due to enhanced downwelling and consistent with recent studies (Neu et al., 2014; Banerjee et al., 2016; Meul et al., 2018). In addition, the positive $O_3$ anomalies induced by the ENSO signal in the extratropics indicate hemispheric asymmetry in both simulations and observations, with a generally weaker response in the southern hemisphere than in the northern hemisphere (Fig. 3(a-c)). This hemispheric asymmetry results from a weak quasi-horizontal mixing between tropics and extratropics induced by the asymmetry in the wave breaking response to El Niño-like conditions, which will be discussed

further in Section 5. The negative $O_3$ anomalies seen in the southern hemisphere polar region reflect the large variability at high latitudes in Antarctic ozone due to chemical $O_3$ loss (Solomon, 1999; Laube et al., 2014; WMO, 2014; Montzka et al., 2018). Note that the absence of $O_3$ anomalies above $500\,\mathrm{K}$ in CLaMS (Fig. 3a) results from the upper boundary condition, which is imposed above this level based on mean climatological fields and thus precludes representation of variability.

Despite generally good agreement between simulations and observations, the signal in the MLS data is weaker in both the

tropics and the extratropics, particularly in the southern hemisphere. The stratospheric entry value of fire emission markers is strongly enhanced under El Niño conditions (Fromm and Servranckx, 2003; Fromm et al., 2006; Trentmann et al., 2006). Increased upper tropospheric $O_3$ mixing ratios have also been linked to increased $O_3$ precursor emissions from biomass burning. The dynamical changes, severe drought, and ensuing large-scale forest fires in Indonesia and Malaysia induced by strong El Niño events have been conclusively associated with substantial anomalies in UTLS CO and $O_3$ (e.g. Thompson et al., 2001;

Logan et al., 2008; Chandra et al., 2007, 2009; Nassar et al., 2009; Livesey et al., 2013; Field et al., 2016). Thus, tropical upper tropospheric $O_3$ mixing ratios may be enhanced not only because of increased convective transport but also increased fire emissions (this was especially true in 2015, Field et al., 2016). Hence, the tropical UTLS $O_3$ mixing ratios will reflect the net change from competing effects that CLaMS simulations cannot capture because of the use of zero $O_3$ as a lower boundary condition, as mentioned earlier. In some cases these local/regional effects may have been large enough to impact the tropical

mean $O_3$ mixing ratios. In the CLaMS model this chemical relationship between CO and $O_3$ is missing, which might explain the discrepancies with MLS observations.

## 4 Structural changes in the lower stratospheric BD-circulation

In this section, various diagnostics of the BD-circulation strength (e.g., AoA, $\overline{w^*}$, $\Psi$, RCTT, age spectrum) from simulations with CLaMS, driven by ERA-I and JRA-55 reanalyses, are analyzed for ENSO-related variability and consistency with the $O_3$-based results (see Sect. 3). In contrast to the complex chemistry in trace gases, the AoA is particularly useful as a diagnostic for investigating variability in stratospheric transport and mixing, as it is not influenced by chemistry.

Figures 4(a-b) show the amplitude variation of the ENSO impact on AoA for the 1979–2013 period. The vertical structure of AoA anomalies depicts a pattern of changes similar to the ENSO imprint on $O_3$ mixing ratios. Negative AoA anomalies (young AoA) emerge throughout the tropics in both ERA-I and JRA-55 reanalyses and propagate upwards into the stratosphere during El Niño-like conditions. Positive AoA anomalies (old AoA) arise in the extratropics with a strong effect in the northern hemisphere during El Niño, leading to hemispheric asymmetry consistent with the $O_3$ anomalies. The picture of AoA anomalies agrees well with $O_3$ anomalies from CLaMS simulations and MLS observations, albeit with a smoother pattern of changes for MLS $O_3$ anomalies in the southern hemisphere lower stratosphere (Fig. 3).

Figures 4(c, d) depict the ENSO-induced variability in the $\overline{w^*}$, indicating a clear increase in the tropical upwelling during El Niño-like conditions, consistent with recent findings (Randel et al., 2009; Calvo et al., 2010; Konopka et al., 2016). The vertical structure of AoA and $O_3$ changes in the UTLS, i.e., negative anomalies in the tropics and positive anomalies in the extratropics during El Niño-like conditions, is mainly explained by the ENSO-induced anomalies in $\overline{w^*}$ and in the diabatic heating rate ($\dot{\Theta}$) (Fig. 4(c-f)). During El Niño, the increase of the ascending branch of the BD-circulation (positive tropical $\overline{w^*}$ and $\dot{\Theta}$) anomalies enhances upward transport of young tropospheric air poor in $O_3$ into the tropical stratosphere. The enhanced downwelling in the mid and high latitudes transports more old stratospheric air rich in $O_3$ downwards into the polar regions (see Fig. 3), contrasting with model projections of shorter stratospheric residence time due to enhanced downwelling in a warming climate (e.g. McLandress and Shepherd, 2009; Lin and Fu, 2013; Butchart, 2014; Hardiman et al., 2014). The main difference in the response of the AoA to El Niño compared to its global warming response lies in the difference in the transition branch response and the difference in time-scale of the El Niño perturbations compared to those induced by a globally warming climate, which is on the order of years. In a warming climate, climate models predict a globally decreasing AoA due to faster upwelling and downwelling of all branches (transition, shallow and deep) over a time-scale of decades, leading to a shorter stratospheric residence time of air parcel tropically ascending. In contrast, during El Niño, the shallow and transition branch evolve in different regime i.e. a weakening transition branch, strengthening shallow branch and not clear response for the deep branch. El Niño strengthening the downwelling of the shallow branch has a typical time-scale of a few months and maximizes in winter, transporting much older air downward to the lower extratropical stratosphere and hence increasing AoA. The El Niño effect is analogous to the effect of seasonality, where also stronger winter downwelling is related to increasing AoA in the extratropical lower stratosphere. Consequently, during El Niño the enhanced tropical upwelling depletes $O_3$ in

the tropical lower stratosphere, while the strengthened downwelling of the shallow branch enhances $O_3$ in the mid and high latitudes. Opposite changes occur during La Niña (not shown). The ENSO-induced variations in $\overline{w^*}$ and $\dot{\Theta}$ agree well in the two reanalyses in terms of morphology, though not in magnitude (see Fig. 4(c-f)). The $\overline{w^*}$ and $\dot{\Theta}$ changes related to El Niño for JRA-55 are more confined in the tropics and exhibit stronger downwelling in the northern hemisphere than those from ERA-I. The latter also exhibits a stronger $\overline{w^*}$ anomalies in the tropics than JRA-55, consistent with the differences between the two reanalyses in $O_3$ anomalies (Fig. 3).

However, as the AoA is affected by both residual circulation and mixing processes (e.g., Garny et al., 2014; Ploeger et al., 2015b, a), there could be an ambiguous relation between AoA changes and upwelling or downwelling. Therefore, we also analyse the ENSO-induced RCTT and $\psi^*$ anomalies (Fig. 5(a–d)). The ENSO impact on the vertical structure of the BD-circulation becomes evident from the mass stream function and the RCTT, i.e. the time scale of transport by the pure residual circulation (Fig. 5). In the tropics, El Niño causes decreasing RCTT throughout most parts of the stratosphere (below 550 K) related to the strengthening tropical residual circulation cell associated with the shallow branch of the BD-circulation (Birner and Bönisch, 2011). The strengthening tropical residual circulation in Fig. 5(a, b) is consistent with positive (negative) stream function changes in the northern hemisphere (southern hemisphere), indicating a strengthening residual mean mass circulation in the tropics (Fig. 5(c, d)). In the extratropical lower stratosphere at altitudes below 450 K, the RCTT increases during El Niño, consistent with a weakening extratropical residual circulation cell related to the transition branch of the BD-circulation (Lin and Fu, 2013). These changes in extratropical RCTT also corroborate a weakening of residual circulation cells in the extratropics of both hemispheres during El Niño. The pattern of changes in the residual circulation (transit time and stream function) depicts a weakening transition branch during El Niño, while the shallow branch is strengthening in both reanalyses. However, differences occur between the two reanalyses concerning the strength of the shallow branch. The strengthening of the shallow branch in response to El Niño does not extend as far poleward in JRA-55 as it does in ERA-I, reflecting the difference in the strength of the tropical upwelling response in the two reanalyses (Fig. 4(c, d)). ENSO-induced variability in the deep branch is less evident in the reanalyses (not shown).

Next we quantify the changes in the strength of the transition and shallow branches. Figure 6(a-d) shows the lag-correlation of the ENSO-induced changes in the transition and shallow circulation branches inferred from the RCTT anomalies versus the MEI. A lag-correlation is calculated for each given latitude and altitude grid point in these two regions: 20–60° and 370–420 K for the transition branch and 10–60° and 420–500 K for the shallow branch. Note that positive lag-correlations imply weaker circulation, and negative ones imply stronger circulation. The estimated changes in the transition and shallow branches are as larger as $\pm 8\%$ over the 1979-2013 period, except the strong Niño in 1997 where the changes in the shallow branch from ERA-I reach $-10\%$. These changes are robust as shown by the lag-correlation estimated from the transition branch versus MEI, which reaches 73% for ERA-I and 75% for JRA-55. For the shallow branch, the lag-correlation is $-73\%$ for ERA-I and $-53\%$ for JRA-55. The vertical structure of changes in the BD-circulation during El Niño with a strengthening ascending branch and a weakening circulation in the midlatitudes lower stratosphere, is also consistent with the strengthening shallow circulation branch and a weakening transition branch. The vertical structure of the BD-circulation branches agrees between the two reanalyses, although the changes in JRA-55 are more confined in latitude and altitude than changes in ERA-I, consistent

with the variations in diabatic heating rates related to ENSO in the two reanalyses (Fig. 4(e, f)) as well as with the differences in the lag-correlation.

The most complete transit time diagnostic is the age spectrum, which includes the full transit time information related to all circulation pathways and mixing processes, thereby giving clearer insight into the reanalysis differences. From Figures 7(a–d), it can be concluded that the ENSO-induced variations in the age spectrum appear to be mainly caused by changes in the residual circulation and mass stream function (Fig. 5). The El Niño and La Niña impacts on the fraction of young air masses in the tropics and extratropics are consistent with the structural changes in the residual circulation induced by ENSO. Both reanalyses show an increase of the fraction of young air masses with age shorter than about 6 months during El Niño and a significant decrease during La Niña in the tropical lower stratosphere (here 10°S–10°N at 400 K) (Fig. 7(a, b)). Note that JRA-55 depicts a smaller El Niño impact on the youngest air mass fraction than ERA-I, consistent with the reanalysis differences in the RCTTs (Fig. 5). The age spectrum tail, which is most sensitive to changes in mixing with very old air from the extratropics, is unchanged in both reanalyses after 25 months. Hence, the ENSO-induced changes in the tropical age spectrum mainly reflect the strengthening upwelling branch of the residual circulation in the tropics during El Niño, in agreement with the discussion by Konopka et al. (2016). In the lower stratosphere at midlatitudes (here 40°–55°N at 350 K), the age spectrum shows a decrease in the fraction of young air and a slight change in the spectrum tail after 40 months during El Niño, indicating a long-lasting ENSO signal in the northern hemisphere and mixing effects (Fig. 7 (c, d)). The amplitude of decreasing young air mass fraction during El Nino is larger in JRA-55 than in ERA-I, corroborating the observed differences in the AoA and RCTTs (Fig. 4 and 5). These changes in the age spectrum are consistent with the weakening transition branch of the residual circulation in the lowermost stratosphere at midlatitudes (see Fig. 5 and related discussion). Hence, the ENSO-induced changes in the lower stratospheric age spectra are consistent with the structural changes in the residual circulation, with El Niño causing an upward shift of the poleward outflow from the shallow branch of the BD-circulation and a weakening of the transition branch below. La Niña causes the opposite changes.

A very clear picture of the structural circulation changes induced by ENSO emerges from the separation of the young air mass fraction with transit time shorter than 6 months (Fig. 8(a, b)) and the old air mass fraction with transit time longer than 24 months (Fig. 8(c, d)), calculated from the age spectrum. During El Niño, the young air mass fraction with transit time shorter than 6 months increases throughout the tropical lower stratosphere and extends poleward in the layer between about 400–500 K. These changes in young air mass fraction are consistent with a strengthened shallow branch. In contrast, below about 400 K, the poleward transport of young tropical air weakens, and a negative young air anomaly even occurs during El Niño, consistent with the weakening transition branch and isolated midlatitudinal regions. Hence, El Niño clearly strengthens the shallow branch of the BD-circulation (420-500 K) and weakens the transition branch in both reanalyses, with a hemispheric asymmetry. The ENSO-induced variations in the air mass fraction with transit time longer than 24 months consistently show a significant decrease in the tropics and a significant increase in the extratropics in both reanalyses during El Niño (Fig. 8(c, d)). Differences between ERA-I and JRA-55 reanalyses are larger in the old air mass fractions, especially in the extratropics above 400 K, where JRA-55 exhibits larger positive anomalies in older air mass fraction than ERA-I. The signal of the old air mass fraction with transit time longer than 24 months from JRA-55 spreads throughout the lower stratosphere except in

the tropics. Despite the differences in the distribution of the old air mass fraction between ERA-I and JRA-55 reanalyses, the decrease of old air in the tropics and the increase of old air in the extratropics is consistent between the two reanalyses. Note that the ENSO-induced changes are less evident above about 600 K (not shown), indicating that the ENSO impact on the BD-circulation is largely confined to the region below and hence to the transition and shallow branches.

## 5   Discussion

In a recent study, Yang et al. (2014) showed from an idealized model that zonally symmetric SST perturbations drive the deep branch of the stratospheric BD-circulation, whereas zonally localized SST perturbations drive the shallow circulation branch. Here, we find no clear evidence for an El Niño effect on the deep branch of the BD-circulation above about 600 K. Nevertheless, our results are consistent with the findings of Yang et al. (2014), who suggested that a zonally symmetric anomalous SST pattern
like during El Nino, strengthens the shallow branch of the BD-circulation and suppresses the isentropic mixing induced by a stronger subtropical jet. Furthermore, we found evidence that El Niño alters the two sub-branches of the BD-circulation i.e., strengthens the shallow branch between about 420–500 K and weakens the transition branch between about 370–420 K. The strengthening of the deep branch related to El Nino is less evident in the reanalyses examined here.

Insight into the underlying dynamical mechanism causing the changes in the transition and shallow branches of the BD-
circulation is derived from the temperature, zonal mean wind, Eliassen-Palm flux (EP-flux) and EP-flux divergence variations related to ENSO (Fig. 9). Generally both reanalyses agree well in ENSO-induced variations in temperature, zonal mean wind and EP-flux divergence anomalies. In the tropics (30° S–30° N), El Niño clearly warms the upper troposphere and cools the lower stratosphere in both reanalyses, consistent with previous studies (e.g., Randel et al., 2009; Calvo et al., 2010; Simpson et al., 2011). Large tropical temperature changes remain confined below about 500 K. In the extratropics, El Niño generally
warms the whole lower stratosphere, except below about 400 K near the subtropical jets, where negative temperature anomalies occur (Fig. 9(a, d)). The cooling of the tropical stratosphere and warming of the extratropical lower stratosphere are consistent with the increased tropical upwelling and extratropical downwelling during El Niño. In addition, the negative temperature anomalies in the midlatitudes are consistent with a weakening transition branch. The strong differences in the temperature changes between the upper tropical troposphere and the midlatitudes (i.e. strong tropical-midlatitudinal temperature gradient)
cause a strengthening of the subtropical and polar zonal jets on their equatorward flanks, resulting in an equatorward and an upward shift of the subtropical jet (∼10° and ∼10 K) (Fig. 9(b, e)), consistent with the results of Lorenz and DeWeaver (2007). According to Simpson et al. (2011), this equatorward shift of the midlatitude jet related to El Niño results in an enhanced source of waves with higher phase speeds in the midlatitudes and changed propagation characteristics into the stratosphere. Recently, Albers et al. (2018) also attributed the ENSO-related jet variability to wave breaking frequency rather than to the
typical ENSO teleconnection patterns. The temperature and zonal mean wind variations induced by El Niño shown in Fig. 9 agree with prior model and observational studies (e.g., Lu et al., 2008; Simpson et al., 2011; Abalos et al., 2017; Zhou et al., 2018) and with the idealized model results from Yang et al. (2014). This ENSO-induced variability in temperatures and zonal wind can be understood as a direct response to the zonal extent of the SST perturbations.

The changes in EP-flux divergence related to El Niño show positive anomalies at lower levels close to the tropopause and negative anomalies in the midlatitude lower stratosphere above about 420 K (Fig. 9(c, f)). The positive anomalies suggest decreased wave breaking at lower levels in the lower stratosphere during El Niño (McLandress and Shepherd, 2009; Calvo et al., 2010; Simpson et al., 2011), consistent with the weakening of the transition branch. In contrast, the negative anomalies above indicate that more waves break at higher levels in the extratropical lower stratosphere, depositing their momentum flux in these regions, and therefore accelerating the shallow branch (Fig. 9(c, f)). Hence, the wave drag changes shown in Fig. 9(c, f) are qualitatively consistent with a weakening of the transition branch of the BD-circulation and a strengthening of the shallow branch during El Niño. These wave drag changes are also consistent with the findings of Zhou et al. (2018), who concluded that the magnitudes of the stratospheric zonal mean responses are larger in the case of extreme El Niño events, as the strong upward propagation of planetary-scale waves induces a weaker northern hemisphere polar vortex by breaking at high latitudes.

Gravity waves have been shown to play an important role in driving ENSO related variations in the lower stratospheric circulation, particularly in the subtropics (Calvo et al., 2010; Simpson et al., 2011). According to Alexander et al. (2017), zonal gravity wave momentum fluxes at the tropopause were 11 % smaller during El Niño than during La Niña because of a shift in the precipitation to the central Pacific, where upper tropospheric zonal winds are less favorable for vertical wave propagation. According to Kawatani et al. (2010), zonal mean variation of wave forcings in the stratosphere results from the phase of the QBO and the changes in wave sources, i.e. the vertical shear of zonal mean winds associated with the Walker circulation. Hence, close agreement between the ENSO variations in the wave drag and the residual circulation variations is not necessarily expected, due to the strong effect of gravity waves.

To quantify the contribution of wave drag to the changes in the transition and shallow branches of the BD-circulation induced by El Niño, the zonal mean wave drag of the explicitly resolved waves (both global-scale and gravity waves) is calculated from the divergence of the EP-flux using the ERA-I. The estimate for the planetary wave drag is then obtained by integrating the EP flux divergence over zonal wave numbers 1-20. According to Alexander and Rosenlof (1996), the missing wave drag in ERA-I can be assumed to be the part of the contribution of gravity wave drag in the zonal mean momentum budget that is not explicitly resolved by the model grid, and its relative variations should still contain valuable information. The total gravity wave drag is estimated as the sum of the missing drag and the model-resolved waves integrated over zonal wave numbers 21-180. Fore more details about the calculations and inter-comparisons of the ERA-I wave drag with those derived from satellite observations see Ern et al. (2014, 2015, 2016).

Figure 10(a-c) shows the zonal mean distribution of the ENSO impact on monthly-mean net wave forcings (PWD+GWD-du/dt) (a), planetary wave drag (PWD) (b) and gravity wave drag (GWD) (c). The net wave forcings (Fig. 10a) explain the changes in the branches and the hemispheric asymmetry to a remarkable degree. Clearly, the weakening of the transition branch is due to an upward shift in the wave dissipation height up to 425 K, while the strengthening of the shallow branch results from wave breaking above 425 K. The hemispheric asymmetry is a consequence of the asymmetry in both wave distributions (global-scale and gravity), with a larger contribution in the northern hemisphere than southern hemisphere. Most of the ENSO-induced variations in wave forcing are contained in the zonal wavenumbers up to 20 (global-scale waves) and are focused around the tropopause. In the northern hemisphere, there is a positive pattern of planetary wave changes above the tropopause and

a negative pattern below the tropopause over a wide latitude range in the extratropics (Fig. 10b), consistent with results from the WACCM model (see Fig. 3 of Calvo et al., 2010). This pattern of changes indicates an altitude shift in the dissipation height of the global-scale waves. In the southern hemisphere, the pattern of planetary wave changes is somewhat different and indicates a general shift towards positive values. For the gravity wave response to El Niño, Fig. 10c shows a positive response

in the subtropics around 380 K, i.e. a reduction in wave drag, which is however weaker than the planetary wave response. Interestingly, there is a negative response at higher altitudes in the northern hemisphere subtropics between 425 and 550 K, i.e. an increase in wave drag, that is even stronger than the response for the zonal wavenumbers up to 20 (Fig. 10b). In summary, the altitude shift in the dissipation height of the large-scale and gravity waves clearly causes the slowdown of the transition branch, while the gravity wave breaking in the tropics-subtropics combined with planetary wave breaking at high latitudes drive the

acceleration of the shallow branch. Gravity wave breaking in the subtropics close to the edge of the upwelling region contributes the most to the strengthening of the tropical upwelling. Driven by the wave breaking, the mixing efficiency between tropics and extratropics will be different in the northern and southern hemispheres, leading to the observed hemispheric asymmetry. In addition to the lack of tropospheric $O_3$ chemistry and the $O_3$ lower boundary condition set to zero in CLaMS. Uncertainties in the upwelling strength also contributes to the factor of 2 difference observed in the $O_3$ anomalies (Fig. 1-3).

Future projections of climate models predict a shift of the ENSO basic state toward more frequent El Niño conditions in a warming climate due to an increase in anthropogenic greenhouse gases (Timmermann et al., 1999; van Oldenborgh et al., 2005; Latif and Keenlyside, 2009; Cai et al., 2014). As changes in UTLS trace gases, including $O_3$ and $H_2O$ (Solomon et al., 2010; Riese et al., 2012), directly impact the global radiative forcing of climate (Forster and Shine, 1999; Butchart and Scaife, 2001), it is crucial to understand such future changes in trace gases induced by a shift of the ENSO basic state toward more frequent

El Niño-like conditions. Despite the uncertainty in the magnitude of the future El Niño events, we speculate that the projected change in the El Niño occurrence frequency will cause structural changes similar to the current $O_3$, RCTT and $\dot{\theta}$ anomalies (Fig. 3, 4 and 5). In a future climate characterised by a shift of the basic state toward more frequent El Niño conditions, the negative $O_3$ anomalies in the tropics and positive $O_3$ anomalies in the midlatitudes will strengthen (by at least 15 %), enhancing stratosphere-to-troposphere of ozone mass flux (Holton et al., 1995; Hegglin and Shepherd, 2009; Zeng et al., 2010; Neu et al.,

2014; Yang et al., 2016; Albers et al., 2018; Meul et al., 2018) and stronger ozone radiative feedback (Forster and Shine, 1997; Birner and Charlesworth, 2017; Ming et al., 2017).

## 6 Summary and Conclusions

Based on an established multiple regression method applied to MLS observations and CLaMS simulations driven by ERA-I and JRA-55 reanalyses, we found that ENSO induces structural changes of the BD-circulation in the lower stratosphere. These

structural changes in the BD-circulation lead to substantial changes in the tropical and midlatitudinal lower stratospheric $O_3$ anomalies of about 15 % for MLS observations with a hemispheric asymmetry (i.e. stronger $O_3$ changes in northern hemisphere than in southern hemisphere). This circulation asymmetry results from the asymmetry in wave breaking response to ENSO.

The regression analysis of different metrics of the circulation strength related to ENSO, including mean AoA, $\overline{w^*}$, RCTT, $\psi^*$ and age spectra, shows structural changes in the lower stratospheric BD-circulation branches, consistent with observed $O_3$ anomalies. The ENSO influence on the BD-circulation turns out to be mainly evident for the transition and shallow circulation branches (Birner and Bönisch, 2011; Lin and Fu, 2013). During El Niño, the transition branch (370–420 K) weakens, while the shallow branch (420–500 K) strengthens. These structural changes in the transition and shallow branches are as large as $\pm 8\%$ and are tightly linked to the dynamical response of the atmosphere to ENSO. During El Niño, the strengthened tropical-midlatitudinal temperature gradient induces a strengthening of the subtropical zonal jets on their equatorward flanks, resulting in an equatorward and upward shift of the subtropical jet. This equatorward shift of the midlatitude jet induced by El Niño results in enhanced wave propagation towards the extratropical lower stratosphere and breaking therein, consistent with the structural changes in the BD-circulation. The decomposition of the wave drag into planetary and gravity wave drag drag led to a quantification of the contributions of these two groups to the weakening transition and strengthening shallow branches. The upward shift in the dissipation height of the large-scale and gravity waves drives the slowdown of the transition branch, while enhanced gravity wave breaking in the tropics-subtropics (above about 425 K) mainly drives the acceleration of the shallow branch combined with a contribution from planetary wave breaking at high latitudes. The contribution of gravity waves mainly predominates in the strengthening of the shallow branch. During La Niña, opposite change occurs (not shown).

These structural circulation changes related to ENSO affect the distributions of radiatively active trace gases in the UTLS, including $O_3$, which, in turn, crucially impact the global radiation budget (Forster and Shine, 1999; Butchart and Scaife, 2001). Hence, the ENSO influence on the structure of the BD-circulation in the UTLS opens a pathway for a stratospheric impact on future climate. It is thus necessary, that climate models represent these processes well to achieve reliable climate projections. Our results suggest that in the context of a changing future climate, where increasing El Niño-like conditions (Timmermann et al., 1999; Cai et al., 2014) and decreasing lower stratospheric QBO amplitude (Kawatani and Hamilton, 2013) are expected, the ENSO effect will be increasingly important for controlling the distributions of radiatively active greenhouse gases in the UTLS.

*Data availability.* The Aura Microwave Limb Sounder product (http://disc.sci.gsfc.nasa.gov/Aura/data-holdings/MLS/index.shtml, last access: 20 November 2018, Livesey et al., 2017; Santee et al., 2017) and ERA-Interim reanalysis data (https://www.ecmwf.int/en/forecasts/datasets/reanalysis-datasets/era-interim, last access: 20 November 2018, Dee et al., 2011) are available. The $O_3$, AoA, $\overline{w^*}$, RCTT and age spectrum data set can be requested from the corresponding author Felix Ploeger (f.ploeger@fz-juelich.de).

*Author contributions.* All co-authors made substantial contributions to the analysis and interpretation of the data as well as contributing to drafting the article.

*Competing interests.* The authors declare that they have no conflict of interest.

*Acknowledgements.* We particularly thank the NASA Jet Propulsion Laboratory, the European Centre for Medium-Range Weather Forecasts and the Japan Meteorological Agency for providing Aura Microwave Limb Sounder product (https://mls.jpl.nasa.gov/), the ERA-Interim and JRA-55 reanalyses data. Work at the Jet Propulsion Laboratory, California Institute of Technology, was done under contract with the National Aeronautics and Space Administration. This work was funded by the Helmholtz Association under grant number VH-NG-1128 (Helmholtz-Hochschul-Nachwuchsforschergruppe), enabling a research stay at the Institute of Energy and Climate Research, Stratosphere (IEK-7), Forschungszentrum in Jülich during which this work was carried out.

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

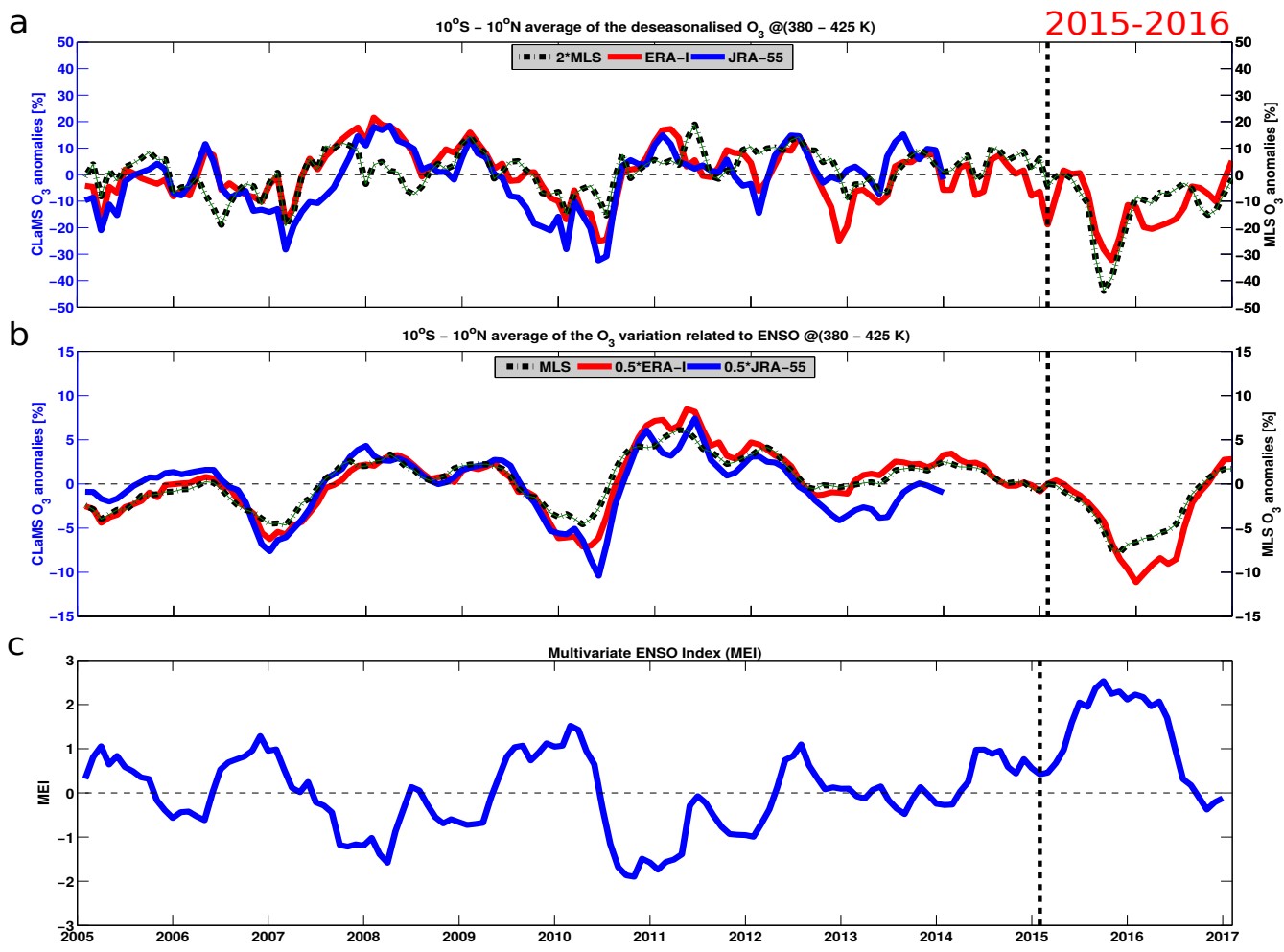

**Figure 1.** Time evolution of the tropical $O_3$ anomalies from CLaMS simulations sampled at the MLS measurement geolocations together with MLS satellite observations in percent change from the monthly zonal mean climatology and averaged between 380–425 K for the 2005–2016 period. Panel (a) shows the $10° S–10° N$ deseasonalised $O_3$ for CLaMS driven by ERA-I (red); CLaMS driven by JRA-55 (blue) and MLS (dashed-black). Panel (b) shows the ENSO-induced $O_3$ anomalies in the tropics for CLaMS driven by ERA-I (red); CLaMS driven by JRA-55 (blue) and MLS (dashed-black) derived from the multiple regression fit. Panel (c) shows the Multivariate ENSO Index (MEI: blue). Note that there is a factor of 2 difference in the legend in Figs. 1(a, b), reflecting the difference in the magnitude of the deseasonalised $O_3$ mixing ratio between CLaMS and MLS. Vertical black dashed line indicates February 2015 for the warm ENSO onset.

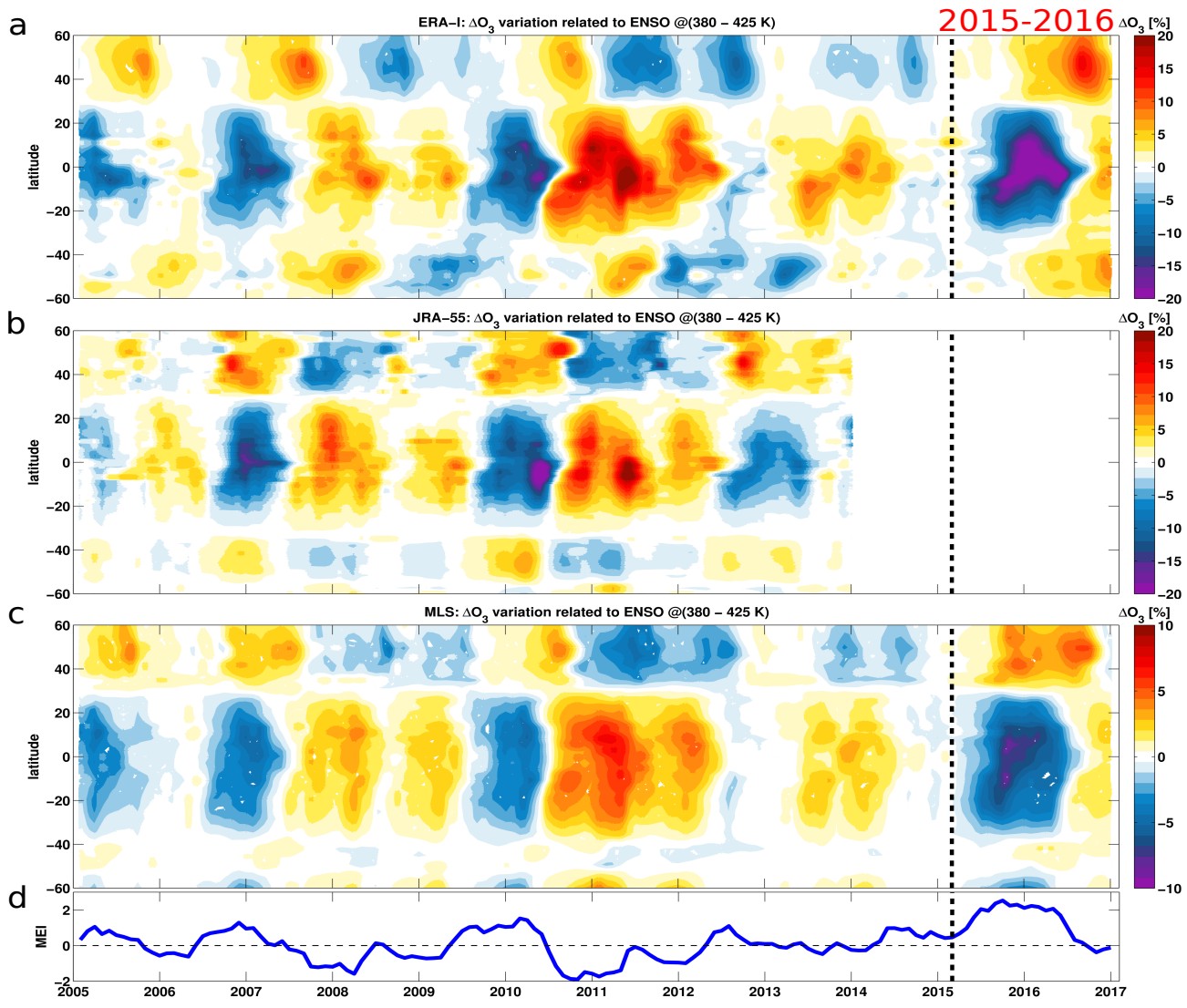

**Figure 2.** Latitude-time evolution of the ENSO impact on lower stratospheric $O_3$ from (a) CLaMS simulations driven by ERA-I; (b) CLaMS simulations driven by JRA-55 and (c) MLS satellite observations in percent change from the monthly zonal mean climatology derived from the multiple regression fit and averaged between 380–425 K for the 2005–2016 period. Note that there is a factor of 2 difference in the color scales in Figs. 2(a, b) and 2c, reflecting the difference in the magnitude of the deseasonalised $O_3$ mixing ratio between CLaMS and MLS. The panel (d) shows the MEI in blue. Vertical black dashed line indicates February 2015 for the warm ENSO onset.

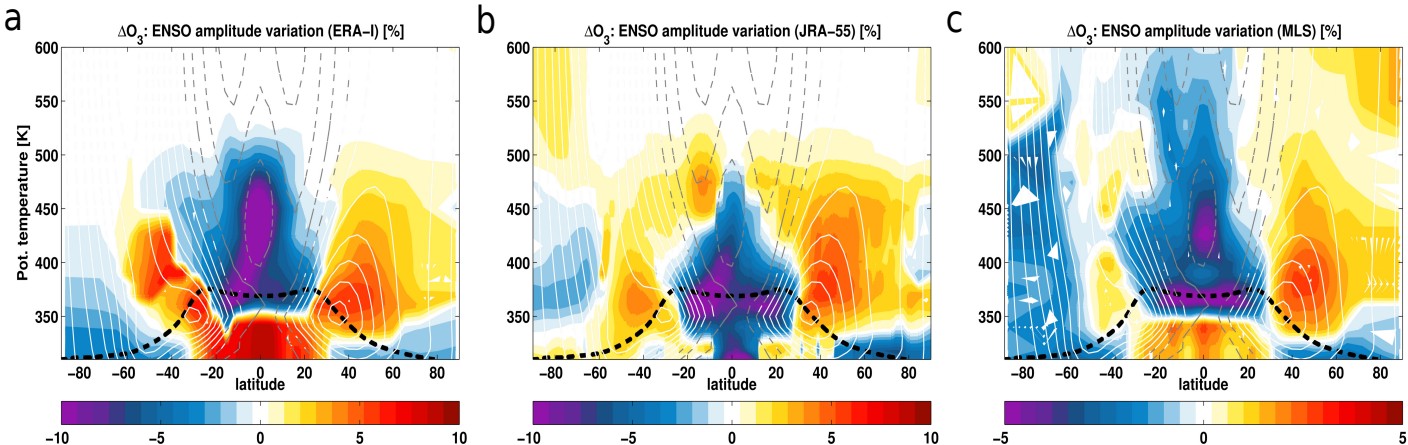

**Figure 3.** Zonal mean distribution of the ENSO impact on stratospheric O$_3$ variability from (a) CLaMS simulations driven by ERA-I; (b) CLaMS simulations driven by JRA-55 and (c) MLS satellite observations in percent change relative to the climatological monthly mean mixing ratios. The amplitude of the O$_3$ variations (term $b_2 \times STD(MEI)$) attributed to ENSO is calculated by projecting the regression fits onto the ENSO basis functions for the 2005–2016 period. Note that there is a factor of 2 difference in the color scales in Figs. 3(a, b) and 3c, reflecting the difference in the magnitude of the O$_3$ changes between CLaMS and MLS. Black dashed horizontal line indicates the climatological tropopause from ERA-I (a, b) and JRA-55 (c). Zonal mean wind component, u (m·s$^{-1}$), averaged over the 2005–2016 period, from ERA-I is overplotted as solid white (westerly) and dashed grey (easterly) lines.

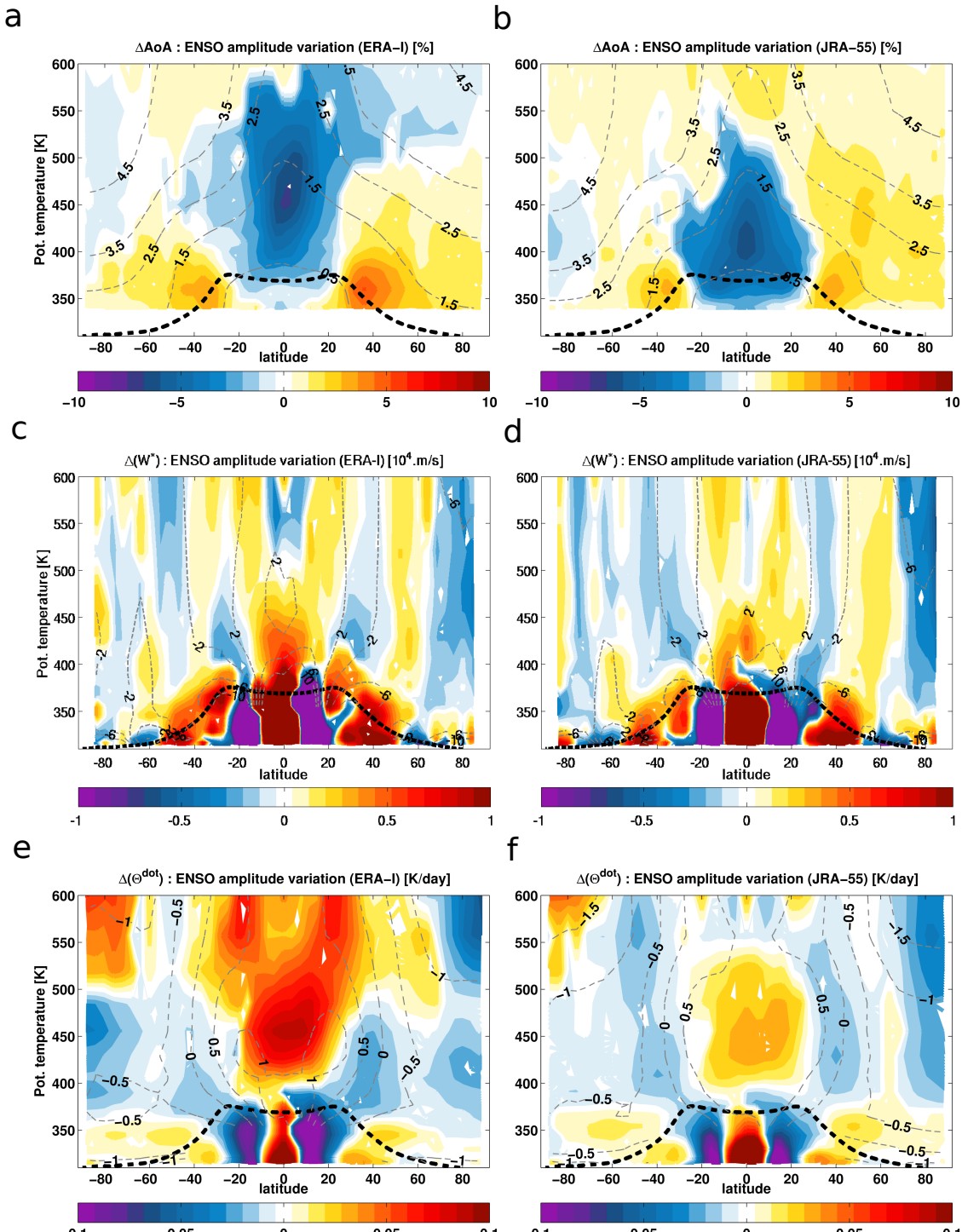

**Figure 4.** Zonal mean distribution of the ENSO impact on mean age (a, b), residual vertical velocity ($\overline{w^*}$) (c, d) and diabatic heating rate ($\dot{\Theta}$) (e, f) from CLaMS simulations driven by ERA-I and JRA-55. The mean age anomalies are in percent change relative to the zonal monthly mean climatology. The units of $\overline{w^*}$ and $\dot{\Theta}$ are in m/s and K/day. The amplitude of the $O_3$ variations (term $b_2 \times STD(MEI)$) attributed to ENSO is calculated by projecting the regression fits onto the ENSO basis functions for the 1979–2013 period. Black dashed horizontal line indicates the climatological tropopause from ERA-I and JRA-55 reanalyses. Zonal mean climatologies of the mean age, $\overline{w^*}$ and $\dot{\Theta}$ are overplotted as

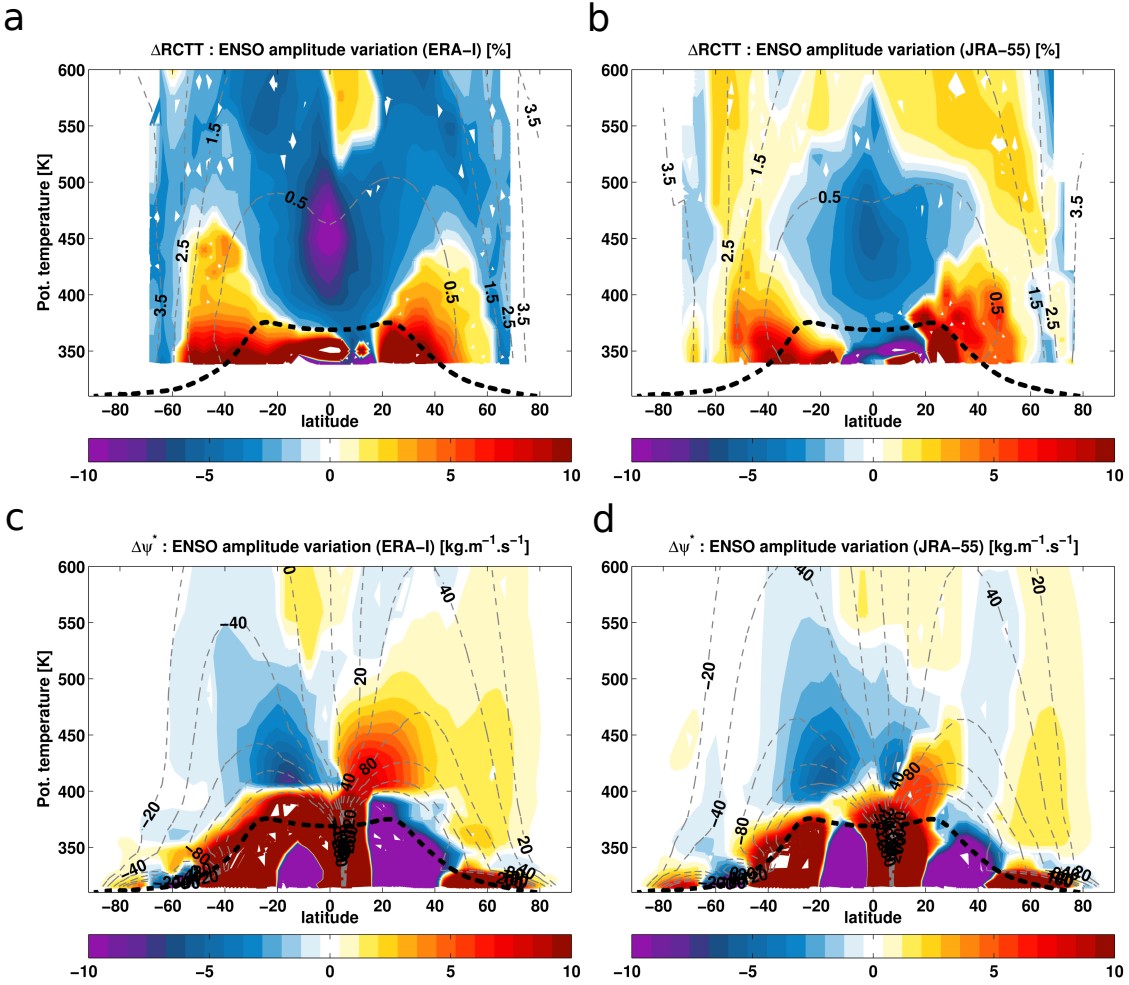

**Figure 5.** Zonal mean distribution of the ENSO impact on residual circulation transit time (RCTT) (a, b) and the residual circulation mass stream function ($\psi^*$) (c, d) from CLaMS simulations driven by ERA-I (a, c) and JRA-55 (b, d). RCTT is shown in percent change relative to the monthly zonal mean climatology. The amplitude of the RCTT and $\psi^*$ variations (term $b_2 \times STD(MEI)$) attributed to the ENSO events is calculated by projecting the regression fits onto the ENSO basis functions for the 1979–2013 period. Black dashed horizontal line indicates the climatological tropopause from ERA-I and JRA-55 reanalyses. Zonal mean climatologies of the RCTT and $\psi^*$ are overplotted as dashed grey lines.

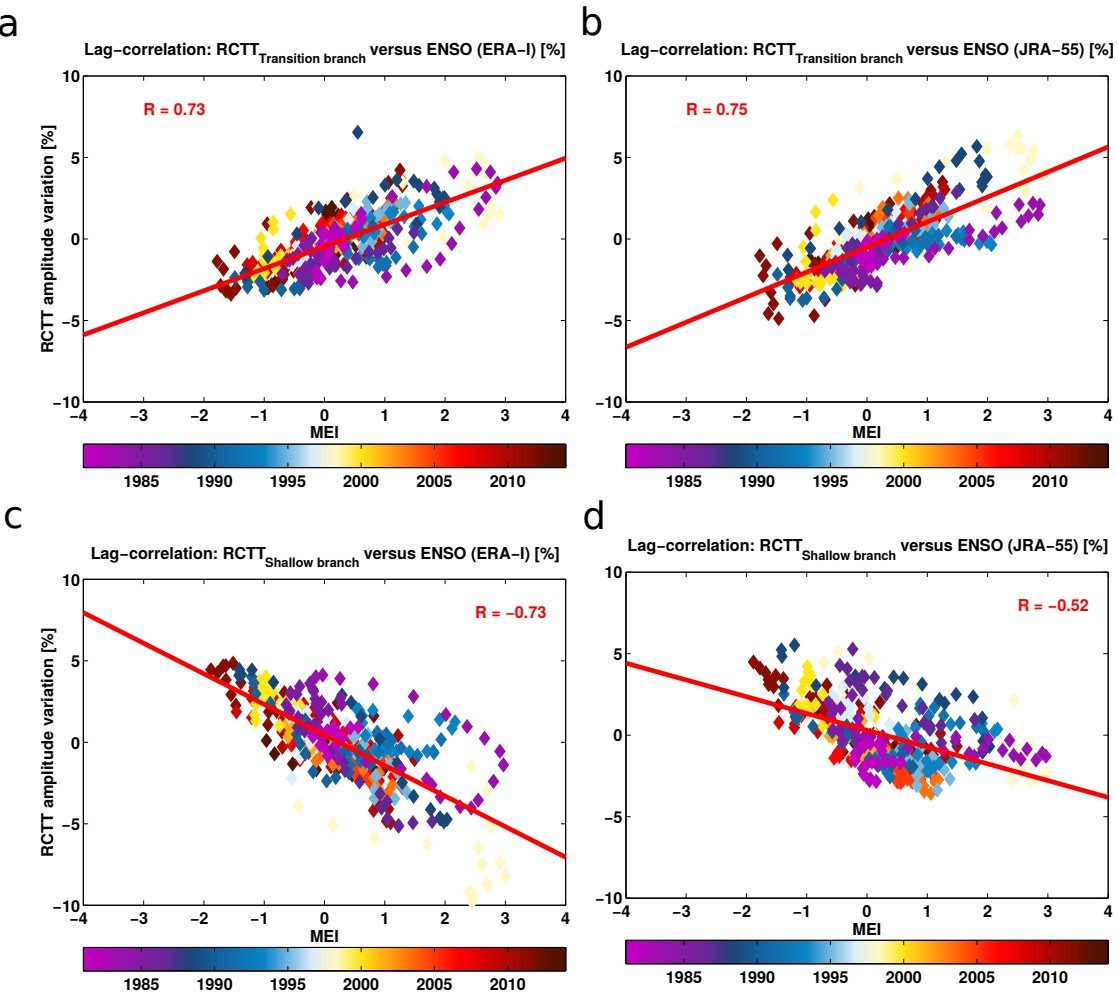

**Figure 6.** Lag-correlation of the ENSO impact on RCTT versus MEI from CLaMS simulations driven by ERA-I (a, c) and JRA-55 (b, d). Transition branch (a, b) and shallow branch (c, d) changes are shown in percent change relative to the monthly zonal mean climatology. The RCTT variations attributed to the ENSO events using the regression analysis are averaged between 20° and 70° and between 370 and 420 K for the transition branch and between 10° and 70° and between 420 and 500 K for the shallow branch during the 1979–2013 period.

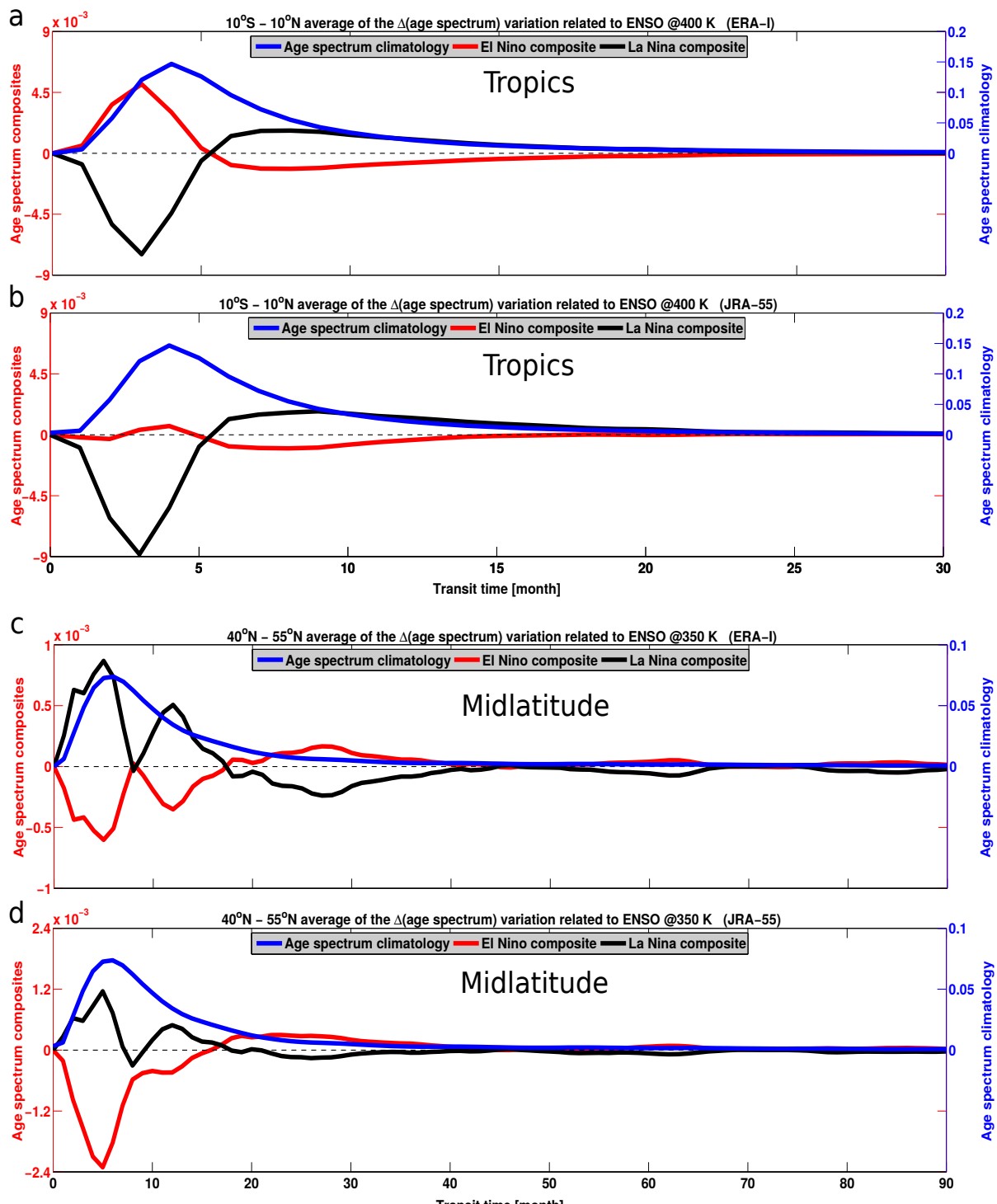

**Figure 7.** ENSO impact on monthly-mean age spectrum from CLaMS simulations driven by ERA-I and JRA-55 reanalyses for the 1979–2013 period: (a, b) tropics at 400 K and (c, d) midlatitudes at 350 K. The El Niño and La Niña composites shown are derived from the multiple regression fit as the difference between the residual ($\varepsilon$ in (1)) without and with explicit inclusion of the ENSO signal. Note that the x-axis for the tropic and midlatitude panels are not the same. The ERA-I and JRA-55 midlatitude panels use different y-axis ranges. The x-axis ranges of the tropical panels stop at 30 months, while the x-axis ranges of the midlatitude panels stop at 90 months.

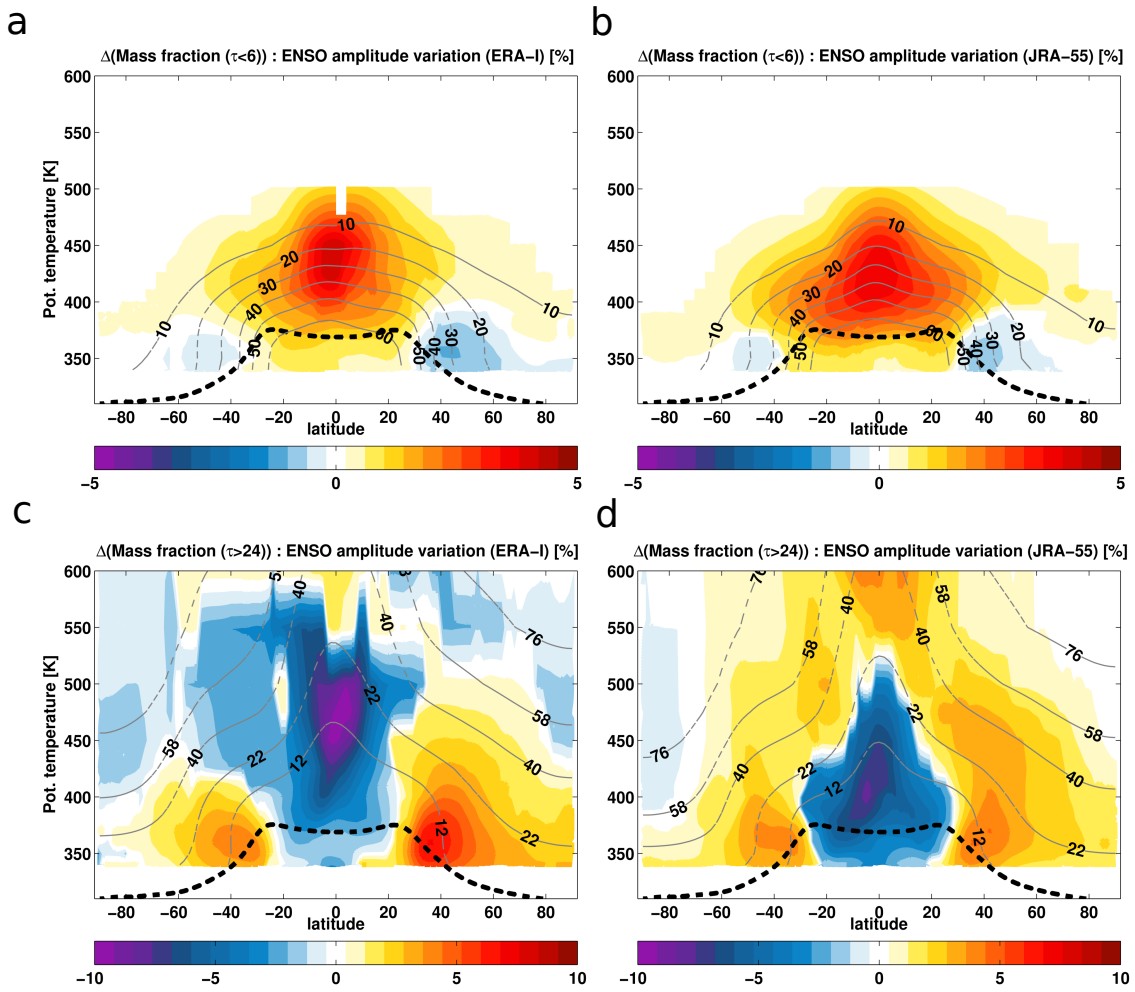

**Figure 8.** Zonal mean distribution of the ENSO impact on monthly-mean young and old air mass fraction from CLaMS simulations driven by (a, c) ERA-I and (b, d) JRA-55 reanalyses. The amplitude of the air mass fraction variations (term $b_2 \times STD(MEI)$) attributed to ENSO is calculated by projecting the regression fits onto the ENSO basis functions for the 1979–2013 period. ENSO amplitude variation of the young air mass fraction with transit time $\tau$ shorter than 6 months is shown in (a, b) panels. ENSO amplitude variation of the old air mass fraction with transit time $\tau$ longer than 24 months is shown in (c, d) panels. Grey contours are the climatology. Black dashed horizontal line indicates the climatological tropopause from ERA-I and JRA-55 reanalyses.

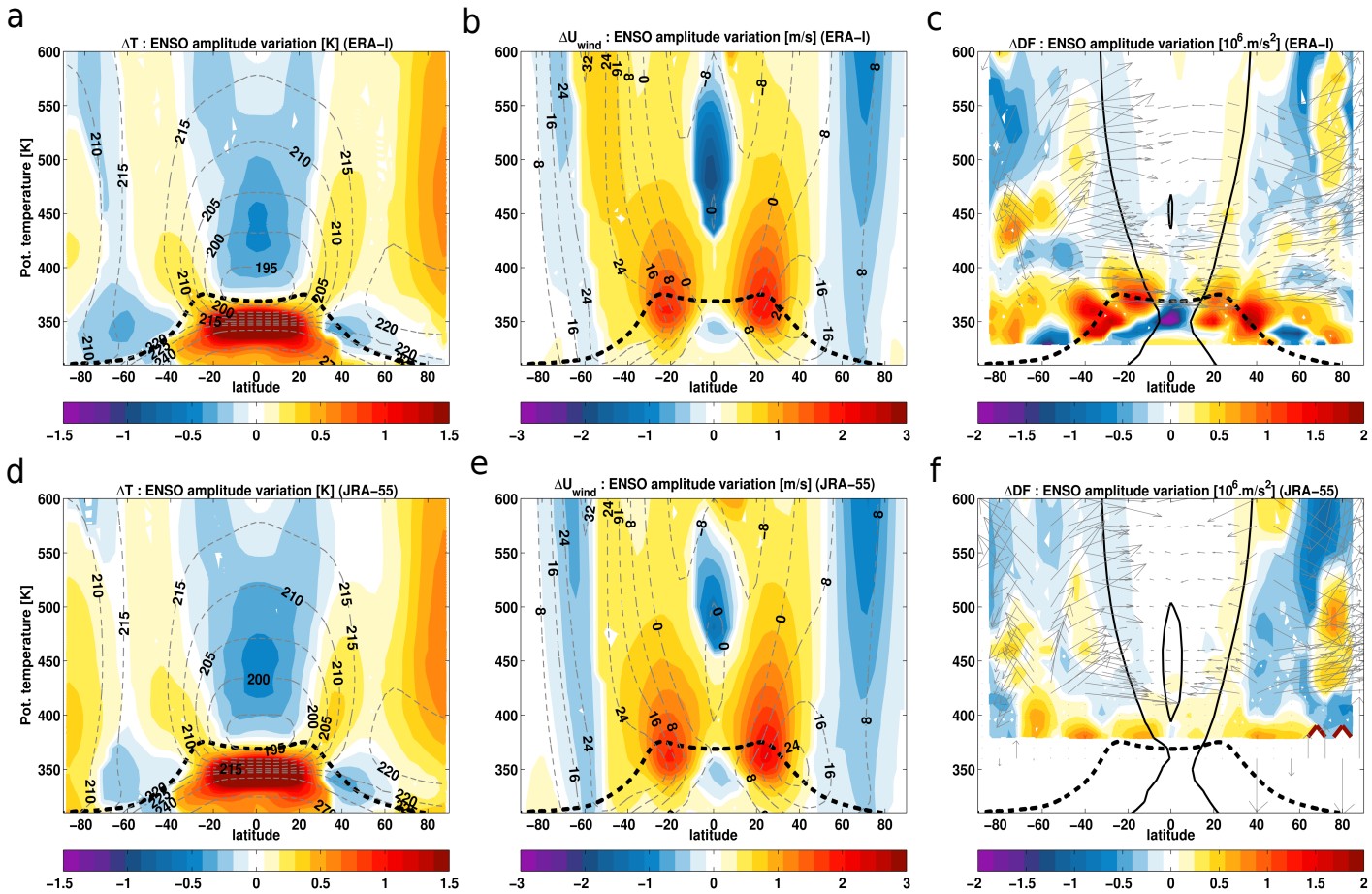

**Figure 9.** Zonal mean distribution of the ENSO impact on monthly-mean temperature [K], zonal wind [m/s] and EP-flux and its divergence [m.s$^{-2}$] derived from (a–c) ERA-I and (d–f) JRA-55 reanalyses. The amplitude of the temperature, zonal wind and EP-flux variations (term $b_2 \times STD(MEI)$) attributed to the ENSO events is calculated by projecting the regression fits onto the ENSO basis functions for the 1979–2013 period. Black dashed horizontal line indicates the climatological tropopause from ERA-I and JRA-55 reanalyses. Zonal mean climatologies are overplotted as dashed grey lines. The thick black line on Fig. 8(c, f) indicates the zero line zonal mean wind. The arrows indicate the EP-Flux vectors.

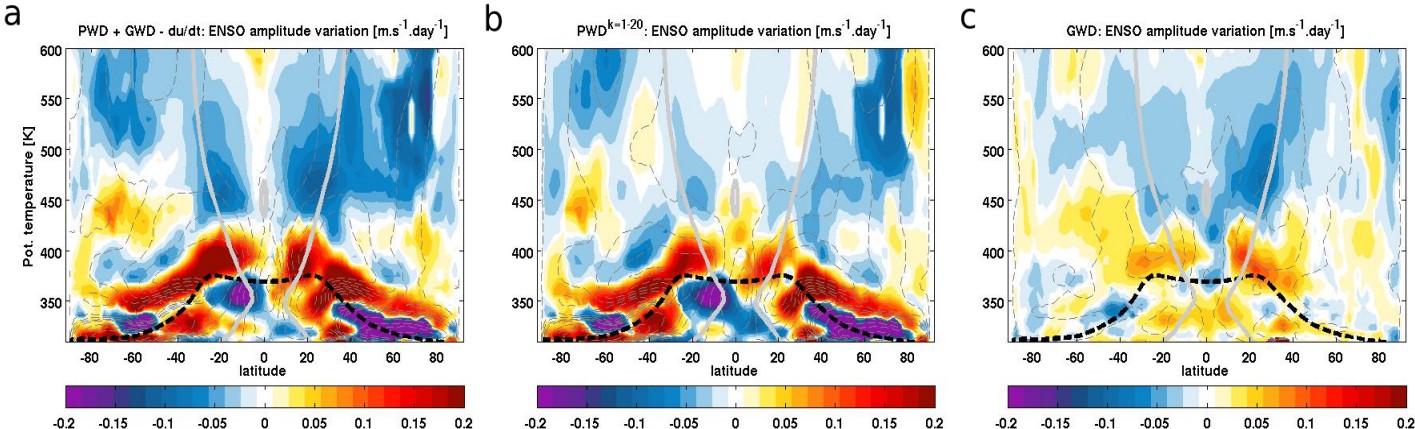

**Figure 10.** Zonal mean distribution of the ENSO impact on monthly-mean net resolved wave drag (a), planetary wave drag (PW) (d) and gravity wave drag (c) derived from ERA-I reanalysis. The amplitude variations (term $b_2 \times STD(MEI)$) attributed to the ENSO events is calculated by projecting the regression fits onto the ENSO basis functions for the 1979–2013 period. Black dashed horizontal line indicates the climatological tropopause from ERA-I. Zonal mean climatologies are overplotted as dashed grey lines. The thick grey line indicates the zero line zonal mean wind.