# Peer review of "Structural changes in the shallow and transition branch of the Brewer-Dobson circulation induced by El Niño"

_Atmospheric Chemistry and Physics, 2018_

## Referee Comment (RC1) · Anonymous Referee #2 · 5 Sep 2018

Review of "Structural changes in the shallow and transition branch of the Brewer-Dobson circulation induced by El Nino" by Diallo et al.

General Comments:

This paper investigates the influences of ENSO on lower stratospheric ozone distribution and BDC strength using a chemical transport model and satellite observations. The ENSO-induced variability in the BDC is thoroughly studied with detailed analysis of the mean age of air, age spectrum, residual circulation, and residual circulation transit time. The authors argue that ENSO causes a structural change of the lower stratospheric BDC, with an increase of the shallow branch and a decrease of the transition branch. Overall the paper is well written. There are some very interesting results. I recommend publication after the authors address my comments.

Specific Comments:

1. The major conclusion of the paper is that ENSO induces structural changes in the BDC. This conclusion needs to be quantified. I suggest that the authors calculate changes in the strength of the transition and shallow branches using the definitions of Lin and Fu (2013).

2. Page 8, line 27-32: These are very interesting results. However, the explanation is not complete. One can also argue that, from the Lagrangian view, an increased downwelling in the extratropics will lead to a decrease of the mean age there, because it takes less time for an air parcel to reach the extratropics. Indeed, many studies have shown that an enhanced downwelling due to global warming is associated with mean age decrease in the extratropics. I am very interested in why the mean age responses to ENSO differ from its responses to global warming.

3. Page 9, line 14-17: Quantify the changes in the transition and shallow branches. See my comment 1.

4. Page 9, line 17-20: Please explain what do you mean by "the upward shift of the strengthening shallow branch".

5. Page 11, line 10-21: The authors appear to suggest that the positive EP flux divergence anomalies near the tropopause are due to the equatorward and upward shift of the jet. Please explain in more detail.

6. Page 11, last paragraph: Is it possible to examine ENSO induced changes in gravity wave drag?

---

## Referee Comment (RC2) · Anonymous Referee #1 · 12 Sep 2018

Reviewer (Comments):
**Review of "Structural changes in the shallow and transition branch of the Brewer-Dobson circulation induced by El Niño" by Mohamadou Diallo et al.**

**Recommendation: Publication after minor revision**

The paper is very well organised and written. The topics discussed in this paper are in general of high relevance and some very interesting results are presented. The conclusions are deduced from comprehensive simulations with a state-of-the-art Lagrangian transport model for the stratosphere (CLaMS) driven by ERA-I and JRA55 in combination with MLS ozone observations and a multi regression model analysis in very well traceable way.

The author pointed out that the key aspect of this study is, that the diagnosed structural changes induced by El Niño (discussed in section 4) are important as they alter key radiative species like ozone in the lower stratosphere (LS) by at least 15% (discussed in section 3) and El Niño-like conditions might be increasing in future.
My main objection is connected with this key aspect which is closely linked to section 3 "El Niño-impact on ozone" and its relation to section 4 "Structural changes in the lower stratospheric BDC". Or more precisely: Has the changes in ozone observed by the MLS satellite really the same morphology than the modelled ozone by CLaMS (driven only by ERA-I) and could these changes be really explained by the changes in transport and dynamical diagnostics derived with the multi regression model from CLaMS simulations driven by both reanalyses? Please, see my general comments.

The paper should be submitted after addressing at least the general comments below.

**General comments:**

The paper pursues two objectives: a) Diagnosing structural changes in the BDC by El Niño using CLaMS simulations driven by two reanalysis datasets, ERA-Interim (ERA-I) and JRA55 respectively, and b) understanding El Niño-induced anomalies in the ozone distribution in the lower stratosphere (LS) as observed by MLS satellite and modelled with CLaMS.
In section 3 or part b), only ERA-I is used for the comparison with ozone MLS observations. This should definitely be done also with JRA55, otherwise it is not possible to interpret the differences between both reanalyses and the relation between El Niño-induced anomalies in transport and dynamical characteristics in terms of observed ozone anomalies in the LS, the climate relevant key aspect (see above).

The part a) is excellently covered in section 4 using CLaMS simulations driven by both reanalyses in combination with a multi regression model decomposing the different parameters for analysing the ENSO-impact on AoA, w*, RCTT, $\Psi$, age spectrum, air mass fraction, temperature (T), zonal mean wind (U), Eliassen-Palm (EP) flux and its divergence.
Both reanalysis datasets show similar morphologies for the different parameters, but also some differences (see specific comments). The El Niño-impact on the parameters listed above elucidate the direct dynamical response (Fig. 8) and the changes in residual (Fig. 4b and 5) and tracer (incl. mixing) transport (Fig. 4a/c, 6 and 7) characteristics in the stratosphere. The overall picture evolving from the analysis of CLaMS simulations driven by ERA-I and JRA55 reanalyses for El Niño events is an increased tropical upwelling, a strengthening and upward

shifting of the subtropical jets. The consequence of this is a strengthening of the shallow branch and a weakening of the transition branch of the BDC during El Niño condition (or episodes). However, the overall picture from section 4 differs in some points from the results in section 3 comparing ozone changes induced by El Niño. The two main differences are:

1.) The magnitude of the ozone changes (anomalies)
2.) The structure of the ozone changes in the SH extratropical LS

To 1.) The author is arguing that the magnitude of the anomalies is biased by the missing tropospheric ozone chemistry and the lower boundary conditions for ozone (set to zero).

To 2.) The author does not discuss in the paper the missing (mainly) positive ozone anomalies in SH extratropics (see Fig. 2 and 3) and the much smoother gradients between tropics and subtropics in the MLS observations. The MLS ozone anomalies are showing a hemispheric asymmetry in the extratropics which is not reflected in ERA-I driven simulations – neither in ozone nor in the transport or dynamical diagnostics in section 4. The positive ozone anomalies in SH extratropics between 350 K and 450 K in CLaMS ERA-I simulation are in line with the anomalies derived in section 4, most obvious in RCTT, and AoA.

What should be at least discussed are other explanations for the differences between observed and modelled distribution of El Niño-induced ozone anomalies. This could be for example:

a) The MLS observations itself, e.g. biases or additional smoothing by resolution and sampling. This issue could be addressed by sampling the model in the way as the satellite is probing the atmosphere.

b) The BDC is not correctly represented in CLaMS driven by ERA-I reanalyses. Hypothesis: The tropical upwelling in and into the LS is too strong or the relation of tropical upwelling and quasi-horizontal (or isentropic) mixing between tropics and extratropics is too weak, especially in SH between 380 and 450K during El Niño. The latter hypothesis would (partially) explain both, the too strong gradients and the too large magnitude of the anomalies.

My last point is that it would be easier for the reader to define the shallow and transition branch in terms of potential temperature (the natural coordinate of CLaMS) in the beginning of this paper and to relate this to the Lin and Fu (2013) paper.

**Specific comments:**

Section 2.1: Please mention the horizontal and vertical resolution of the CLaMS simulations (in the region of interest).

Section 3: The analysis of ozone anomalies related to El Niño derived from CLaMS simulations driven by JRA55 should be added here (see general comments).

P.5, L.13: Releasing the pulses only between 15S and 15N might be biasing the age spectrum results on 350K level in the LMS. It is likely that a significant amount of air originated from outside the tropics (15S-15N) crossing the subtropical jets from the troposphere into the LMS (especially during summer to autumn in the NH), so they are therefore not part of this age spectrum.

P.6, L.25-27: Here, you speculate about the factor of 2 difference between MLS observed and CLaMS ERA-I simulated ozone anomalies. Please see my general comment above. This is important to understand what is driving ozone changes in the LS in order to improve future climate predictions.

P.7, L.12: Description of Fig. 2. Here should be at least mentioned that there are no significant El Niño-induced ozone anomalies in the SH extratropical (30S-60S) MLS observations – in contrast to the CLaMS simulations and to the NH extratropics.

P.7, L30-31: "*In the extratropical UTLS (30°–70°), CLaMS model and MLS observations show a related positive O3 anomaly due to enhanced downwelling.*"
In the LS, MLS observations show only positive ozone anomalies in the NH extratropics (see Fig.3). This is not true for SH extratropics (see general comments). It is true that CLaMS simulations with ERA-I show enhanced downwelling in SH explaining the ozone simulations, but not the observations in the SH.

P.7, L. 35: Could you please explain, why ozone anomalies above 500K are affected by upper boundary conditions and why they are not affected below.

P.8, L13: Again, only missing tropospheric chemistry and lower boundary conditions are to my opinion not sufficient to explain MLS vs. CLaMS-ERA-I differences in Fig. 2 and Fig 3 (see general comments). Or their impact on the ozone anomalies should be quantified somehow.

P.8, L23-25: "*The picture of negative AoA anomalies in the tropical lower stratosphere and positive AoA anomalies in the mid and high latitudes (30°–60° N and S) agrees well with O3 anomalies from CLaMS simulations and MLS observations (Fig. 3).*"
Yes, the picture of AoA and ozone anomalies simulated with CLaMS-ERA-I is consistent, but the picture is not consistent for MLS observations of ozone anomalies in SH extratropical (30-60S) LS.

P.9, L.16-17: "*…, while the shallow branch is strengthening in both reanalyses.*"
Assuming 420 to 550K as the shallow branch, it seems that RCTT and AoA (residual and tracer transport) from JRA55 is not really indicative for a strengthening of the shallow branch. This statement depends strongly on the definition of both branches in potential temperature coordinates (see also general comments).

Figure 6: Please use the same range for the left y-axis for both tropical (and for both midlatitude) plots.

---

## Author Comment (AC1) · 20 Nov 2018

**Answer to Reviewer #1 Comments for "Structural changes in the shallow and transition branch of the Brewer-Dobson circulation induced by El Niño" by Mohamadou Diallo et al.**

Dear Editor-in-Chief, Gabriele Stiller,

We are submitting our revised article titled "Structural changes in the shallow and transition branch of the Brewer-Dobson circulation induced by El Niño". We thank the two Reviewers for their detailed and well thought-out comments, which helped to significantly improve the paper. We have made substantial changes to the manuscript in order to thoroughly address the Reviewers' suggestions and comments. The main changes concern:

- an additional new figure 6 describing the lag-correlation of the ENSO-induced changes in the transition and shallow branches with the MEI in the manuscript as suggested by Reviewer #1 and the related discussion.

- an additional new figure 10 and new paragraph describing the ENSO-induced changes in the net wave forcings, planetary wave drag and gravity wave drag as suggested by Reviewer #1.

- we included Dr. Manfred Ern as as a new co-author, as he provided the wave decomposition data from ERA-I.

- adding two new paragraphs: one for the lag-correlation and another one in the discussion about the decomposition of wave drag as suggested by Reviewer #1.

- recalculation of the ENSO-induced effect on CLaMS O3, which is sampled CLaMS ozone exactly onto MLS locations and discussion of the possible factors that could contribute to the factor of 2 difference between CLaMS and MLS as suggested by Reviewer #2.

- an additional new panel in each of figures 1, 2 and 3 describing the ENSO-induced changes in the CLaMS O3 driven by the JRA-55 reanalysis as suggested by Reviewer #2

- rephrasing of certain paragraphs in order to clarify the manuscript.

With these changes, we are convinced that the paper is highly relevant for a wide-ranging journal like *Atmospheric Chemistry and Physics*. Please see below our answers point by point to all reviewers' comments and suggestions.

Reviewers comments are in bold, followed by our respective replies. Changes in the manuscript are in blue, allowing them to be tracked easily.
Kind regards,
Mohamadou Diallo (on behalf of the co-authors)

**Reviewer #1 (Comments to Author):**

*This paper investigates the influences of ENSO on lower stratospheric ozone distribution and BDC strength using a chemical transport model and satellite observations. The ENSO-induced variability in the BDC is thoroughly studied with detailed analysis of the mean age of air, age spectrum, residual circulation, and residual circulation transit time. The authors argue that ENSO causes a structural change of the lower stratospheric BDC, with an increase of the shallow branch and a decrease of the transition branch. Overall the paper is well written. There are some very interesting results. I recommend publication after the authors address my comments.*

**Specific comments:**

1. ***The major conclusion of the paper is that ENSO induces structural changes in the BDC. This conclusion needs to be quantified. I suggest that the authors calculate changes in the strength of the transition and shallow branches using the definitions of Lin and Fu (2013).***

    We have calculated changes in the strength of the transition and shallow branches from the RCTT as follows. The El Niño induced changes in RCTT for the following regions have been considered: 20–60

degrees and 370-420K for the transition branch and 10–60 degrees and 420-500K for the shallow branch. The lag-correlation of the RCTT anomalies is then inferred with the MEI at each grid point and altitude by taking into account the lag from the regression. Finally, we averaged values over these regions in latitude and altitude bins. This analysis quantifies the ENSO-induced impact on both branches and the associated lag-correlation. These results are shown in the new figure 6 and discussed on Page 10, line 15-27.

We are suggesting here the Simpson et al, (2011) explanation i.e. the equatorward shift of the midlatitude jet induced by El Niño results in an enhanced source of waves with higher phase speeds in the midlatitudes and in changed propagation characteristics into the stratosphere. The positive EP flux divergence anomalies near the tropopause suggest an upward shift in the dissipation level. We have separated the discussion into two paragraphs: one for the jet shift and temperature gradient relationship (Page. 12, line 12-31) another one about the EP-flux and DF (Page 12, line 29-Page 13, line 8).

2. ***Page 8, line 27-32: These are very interesting results. However, the explanation is not complete. One can also argue that, from the Lagrangian view, an increased downwelling in the extratropics will lead to a decrease of the mean age there, because it takes less time for an air parcel to reach the extratropics. Indeed, many studies have shown that an enhanced downwelling due to global warming is associated with mean age decrease in the extratropics. I am very interested in why the mean age responses to ENSO differ from its responses to global warming.***

We thank the Reviewer for pointing that out. The main difference in the response of the AoA to El Niño compared to its global warming response lies in the difference in the transition branch response and, the difference in time-scale of the El Niño perturbations compared to those induced by global warming, which is on the order of years. In a warming climate, climate models predict a globally decreasing AoA due to faster upwelling and downwelling of all branches (transition, shallow and deep) over a time-scale of decades, leading to a shorter stratospheric residence time of air parcel tropically ascending. In contrast, during El Niño, the shallow and transition branch evolve in different regime i.e. a weakening transition branch, strengthening shallow branch and not clear response for the deep branch. El Niño strengthening the downwelling of the shallow branch has a typical time-scale of a few months and maximizes in winter, transporting much older air downward to the lower extratropical stratosphere and hence increasing AoA. The El Niño effect is analogous to the effect of seasonality, where also stronger winter downwelling is related to increasing AoA in the extratropical lower stratosphere. Page 9, line 21-31.

3. ***Page 9, line 14-17: Quantify the changes in the transition and shallow branches. See my comment 1.***

Please see my reply to comment (1).

4. ***Page 9, line 17-20: Please explain what do you mean by "the upward shift of the strengthening shallow branch".***

We have removed the typo "the upward shift of". Page 10, line 30

5. ***Page 11, line 10-21: The authors appear to suggest that the positive EP flux divergence anomalies near the tropopause are due to the equatorward and upward shift of the jet. Please explain in more detail.***

We are suggesting here the explanation by Simpson et al, (2011), i.e. the equatorward shift of the midlatitude jet induced by El Niño results in an enhanced source of waves with higher phase speeds in the midlatitudes and in changed propagation characteristics into the stratosphere. The positive EP flux divergence anomalies near the tropopause and negative changes above suggest an upward shift in the dissipation level. We have separated the discussion into two paragraphs: one for the jet shift and temperature gradient relationship (Page. 12, line 9-28) another one about the EP-flux and DF (Page 12, line 29-end). The modified explanation should be much clearer.

6. ***Page 11, last paragraph: Is it possible to examine ENSO induced changes in gravity wave drag?***

Thanks for this very good suggestion. To address the El Niño impact on the wave drags we have decomposed the zonal mean wave drag of the resolved waves into gravity and large-scale wave drag for ERA-Interim (an analogous analysis for JRA-55 was not possible as we don't have all necessary data available). Figure 10(a-c) shows the zonal mean distribution of the ENSO impact on monthly-mean net wave forcings (PWD+GWD-du/dt) (a), planetary wave drag (PWD) (b) and gravity wave drag (GWD) (c). The net wave forcings (Fig. 10a) explain the changes in the branches and the hemispheric asymmetry to a remarkable degree. Clearly, the weakening of the transition branch is due to an upward shift in

the wave dissipation height up to 425 K, while the strengthening of the shallow branch results from wave breaking above 425 K. The hemispheric asymmetry is a consequence of the asymmetry in both wave distributions (large-scale and gravity), with a larger contribution in the northern hemisphere than southern hemisphere. Most of the ENSO-induced variations in wave forcings are contained in the zonal wavenumbers up to 20 (large-scale waves) and are focused around the tropopause. In the northern hemisphere, there is a positive pattern of planetary wave changes above the tropopause and a negative pattern below the tropopause over a wide latitude range in the extratropics (Fig. 10b), consistent with results from the WACCM model [see Fig. 3 of Calvo et al. (2010)]. This pattern of changes indicates an altitude shift in the dissipation height of the large-scale waves. In the southern hemisphere, the pattern of planetary wave changes is somewhat different and indicates a general shift towards positive values. For the gravity wave response to El Niño, Fig. 10c shows a positive response in the subtropics around 380 K, which is however weaker than the planetary wave response. Interestingly, there is a negative response at higher altitudes in the northern hemisphere subtropics between 425 and 550 K that is even stronger than the response for the zonal wavenumbers up to 20 (Fig. 10b). In summary, the altitude shift in the dissipation height of the large-scale and gravity waves clearly induces the slowdown of the transition branch, while the gravity wave breaking in the tropics-subtropics combined with planetary wave breaking at high latitudes drive the acceleration of the shallow branch. Gravity wave breaking contributes the most to the strengthening of the shallow branch. As gravity wave drag changes occur in the subtropics close to the edge of the upwelling region, they are likely more effective in driving the structural circulation changes than planetary waves. We added a thorough discussion of these new results in Page 13, line 17-35, Page 14, line 1-12.

---

## Author Comment (AC2) · 20 Nov 2018

Please see the enclosed file

Please also note the supplement to this comment:
https://www.atmos-chem-phys-discuss.net/acp-2018-688/acp-2018-688-AC2-supplement.pdf

—————————————

---

## Author Comment (AC3) · 20 Nov 2018

**Answer to Reviewer #2 Comments for "Structural changes in the shallow and transition branch of the Brewer-Dobson circulation induced by El Niño" by Mohamadou Diallo et al.**

Dear Editor-in-Chief, Gabriele Stiller,

We are submitting our revised article titled "Structural changes in the shallow and transition branch of the Brewer-Dobson circulation induced by El Niño". We thank the two Reviewers for their detailed and well thought-out comments, which helped to significantly improve the paper. We have made substantial changes to the manuscript in order to thoroughly address the Reviewers' suggestions and comments. The main changes concern:

- an additional new figure 6 describing the lag-correlation of the ENSO-induced changes in the transition and shallow branches with the MEI in the manuscript as suggested by Reviewer #1 and the related discussion.

- an additional new figure 10 and new paragraph describing the ENSO-induced changes in the net wave forcings, planetary wave drag and gravity wave drag as suggested by Reviewer #1.

- we included Dr. Manfred Ern as as a new co-author, as he provided the wave decomposition data from ERA-I.

- adding two new paragraphs: one for the lag-correlation and another one in the discussion about the decomposition of wave drag as suggested by Reviewer #1.

- recalculation of the ENSO-induced effect on CLaMS O3, which is sampled CLaMS ozone exactly onto MLS locations and discussion of the possible factors that could contribute to the factor of 2 difference between CLaMS and MLS as suggested by Reviewer #2.

- an additional new panel in each of figures 1, 2 and 3 describing the ENSO-induced changes in the CLaMS O3 driven by the JRA-55 reanalysis as suggested by Reviewer #2

- rephrasing of certain paragraphs in order to clarify the manuscript.

With these changes, we are convinced that the paper is highly relevant for a wide-ranging journal like *Atmospheric Chemistry and Physics*. Please see below our answers point by point to all reviewers' comments and suggestions.

Reviewers comments are in bold, followed by our respective replies. Changes in the manuscript are in blue, allowing them to be tracked easily.
Kind regards,
Mohamadou Diallo (on behalf of the co-authors)

**Anonymous Referee #2:**

**The paper is very well organised and written. The topics discussed in this paper are in general of high relevance and some very interesting results are presented. The conclusions are deduced from comprehensive simulations with a state-of-the-art Lagrangian transport model for the stratosphere (CLaMS) driven by ERA-I and JRA55 in combination with MLS ozone observations and a multi regression model analysis in very well traceable way.**
**The author pointed out that the key aspect of this study is, that the diagnosed structural changes induced by El Nio (discussed in section 4) are important as they alter key radiative species like ozone in the lower stratosphere (LS) by at least 15% (discussed in section 3) and El Niño- like conditions might be increasing in future.**
**My main objection is connected with this key aspect which is closely linked to section 3 "El Niño-impact on ozone" and its relation to section 4 "Structural changes in the lower stratospheric BDC". Or more precisely: Has the changes in ozone observed by the MLS satellite really the same morphology than the modelled ozone by CLaMS (driven only by ERA-I) and could these changes be really explained by the changes in transport and dynamical diagnostics derived with the multi regression model from CLaMS simulations driven by both reanalyses? Please, see my general comments.**

**Major comments:**

*The paper pursues two objectives: a) Diagnosing structural changes in the BDC by El Niño using CLaMS simulations driven by two reanalysis datasets, ERA-Interim (ERA-I) and JRA55 respectively, and b) understanding El Niño-induced anomalies in the ozone distribution in the lower stratosphere (LS) as observed by MLS satellite and modelled with CLaMS. In section 3 or part b), only ERA-I is used for the comparison with ozone MLS observations. This should definitely be done also with JRA55, otherwise it is not possible to interpret the differences between both reanalyses and the relation between El Niño-induced anomalies in transport and dynamical characteristics in terms of observed ozone anomalies in the LS, the climate relevant key aspect (see above).*

We have followed the Reviewer suggestions by extending the ozone analysis to JRA-55 even though our CLaMS simulations driven by JRA-55 stop currently at 2013. This extended analysis is very consistent with the ERA-Interim based results. The new JRA-55 results are presented in 3 additional panels and are discussed in section 3.

*The part a) is excellently covered in section 4 using CLaMS simulations driven by both reanalyses in combination with a multi regression model decomposing the different parameters for analysing the ENSO-impact on AoA, $w*$, RCTT, $\psi$, age spectrum, air mass fraction, temperature (T), zonal mean wind (U), Eliassen-Palm (EP) flux and its divergence. Both reanalysis datasets show similar morphologies for the different parameters, but also some differences (see specific comments). The El Niño-impact on the parameters listed above elucidate the direct dynamical response (Fig. 8) and the changes in residual (Fig. 4b and 5) and tracer (incl. mixing) transport (Fig. 4a/c, 6 and 7) characteristics in the stratosphere. The overall picture evolving from the analysis of CLaMS simulations driven by ERA-I and JRA55 reanalyses for El Niño events is an increased tropical upwelling, a strengthening and upward shifting of the subtropical jets. The consequence of this is a strengthening of the shallow branch and a weakening of the transition branch of the BDC during El Niño condition (or episodes). However, the overall picture from section 4 differs in some points from the results in section 3 comparing ozone changes induced by El Niño. The two main differences are:*

*1.) The magnitude of the ozone changes (anomalies)*
*2.) The structure of the ozone changes in the SH extratropical LS*

*To 1.) The author is arguing that the magnitude of the anomalies is biased by the missing tropospheric ozone chemistry and the lower boundary conditions for ozone (set to zero).*

*To 2.) The author does not discuss in the paper the missing (mainly) positive ozone anomalies in SH extratropics (see Fig. 2 and 3) and the much smoother gradients between tropics and subtropics in the MLS observations. The MLS ozone anomalies are showing a hemispheric asymmetry in the extratropics which is not reflected in ERA-I driven simulations – neither in ozone nor in the transport or dynamical diagnostics in section 4. The positive ozone anomalies in SH extratropics between 350 K and 450 K in CLaMS ERA-I simulations are in line with the anomalies derived in section 4, most obvious in RCTT, and AoA.*

*What should be at least discussed are other explanations for the differences between observed and modelled distribution of El Niño-induced ozone anomalies. This could be for example:*

*a) The MLS observations itself, e.g. biases or additional smoothing by resolution and sampling. This issue could be addressed by sampling the model in the way as the satellite is probing the atmosphere.*

We thank the Reviewer for pointing out these issues. We have addressed the Reviewer's concerns by following the suggestions. We have sampled CLaMS ozone exactly at the MLS measurement geolocations as shown in new the Figs. 1-3. The results are still very similar compared to the results from the submitted draft without taking the MLS measurement geolocations into account. Therefore, the difference in the magnitude of the ozone anomalies is not due to a sampling bias.

Concerning the hemispheric asymmetry in the extratropics, both CLaMS driven by ERA-I and JRA-55 together with MLS do show the asymmetry, however, with much smoother gradients between tropics and subtropics of the southern hemisphere in the MLS observations. By introducing the factor 2 of difference in MLS ozone

in Fig. 3c, the asymmetry becomes clearer and comparable to the CLaMS ozone pattern. Note that the ERA-I does show stronger gradient than MLS and JRA-55 partly due to too strong upwelling. As underlined by the Reviewer, the RCTT (transport) does not show the hemispheric asymmetry. However, the age of air (Fig. 4(a,b)), the old air mass fraction (Fig. 8(c,d)), the dynamical diagnostics (T, U in Fig. 9(b, e)) and wave drag (Fig. 10) all do show the hemispheric asymmetry pattern of changes, which is stronger in the northern hemisphere than southern hemisphere in both reanalyses consistent with the ozone anomalies induced by El Niño. The hemispheric asymmetry is likely due to the asymmetry in mixing induced by the wave breaking asymmetry between the two hemispheres (Fig. 10).

*b) The BDC is not correctly represented in CLaMS driven by ERA-I reanalyses. Hypothesis: The tropical upwelling in and into the LS is too strong or the relation of tropical upwelling and quasi-horizontal (or isentropic) mixing between tropics and extratropics is too weak, especially in SH between 380 and 450K during El Niño. The latter hypothesis would (partially) explain both, the too strong gradients and the too large magnitude of the anomalies.*

We thank the Reviewer for these useful comments. We agree with the Reviewer that in addition to the zero boundary condition bias and missing upper tropospheric chemistry, the different factors suggested here can contribute to the stronger magnitude in the ozone anomalies for CLaMS simulations driven by ERA-I. Including the CLaMS ozone driven by JRA-55, the result supports a too strong tropical upwelling in ERA-I, contributing to the observed large magnitude in ERA-I compared to JRA-55 and MLS in the southern hemisphere tropics and subtropics. According to Fig. 10, the southern hemisphere shows a weaker wave breaking than the northern hemisphere, suggesting a weak quasi-horizontal mixing consistent with the El Niño induced effects on aging by mixing (Not shown because a bit noisy). In addition to the difference in the tropical upwelling, the wave breaking asymmetry (mixing) combined with the zero boundary condition and lack of tropospheric chemistry are likely the main reasons that could explain to both, the too strong gradients and the too large magnitude of the anomalies.

*My last point is that it would be easier for the reader to define the shallow and transition branch in terms of potential temperature (the natural coordinate of CLaMS) in the beginning of this paper and to relate this to the Lin and Fu (2013) paper.*

We have rephrased the sentence by defining the shallow and transition branches in potential temperature and then we have related this to the equivalent pressure levels in Lin and Fu (2013). Page 3, line 19-24

**Specific comments:**

1. *Section 2.1: Please mention the horizontal and vertical resolution of the CLaMS simulations (in the region of interest).*

   For the wind and temperature fields, CLaMS uses $1 \times 1$ degree for the horizontal resolution and the native ERA-I and JRA-55 vertical resolution. The mean vertical resolution of air parcels in the CLaMS Lagrangian model is about 400m near the tropopause. Page 4, line 11-13

2. *Section 3: The analysis of ozone anomalies related to El Niño derived from CLaMS simulations driven by JRA-55 should be added here (see general comments).*

   We have done the analysis of ozone anomalies related to El Niño derived from CLaMS simulations driven by JRA-55 and included the results in the manuscript. Please see Fig. 1-3.

3. *P.5, L.13: Releasing the pulses only between 15S and 15N might be biasing the age spectrum results on 350K level in the LMS. It is likely that a significant amount of air originated from outside the tropics (15S-15N) crossing the subtropical jets from the troposphere into the LMS (especially during summer to autumn in the NH), so they are therefore not part of this age spectrum.*

   We agree that pulsing only between 15S and 15N might bias the age spectrum results on 350K level in the LMS. We have commented at page 5, lines 18-21.

4. *P.6, L.25-27: Here, you speculate about the factor of 2 difference between MLS observed and CLaMS ERA-I simulated ozone anomalies. Please see my general comment above. This is important to understand what is driving ozone changes in the LS in order to improve future climate predictions.*

We have rephrased the paragraph (Page 6, line 26-30 and Page 7, line 1-5). The possible factors contributing to the factor of 2 difference between MLS observed and CLaMS ERA-I simulated ozone anomalies are discussed in section 5. Page 14, line 8-13.

5. ***P.7, L.12: Description of Fig. 2. Here should be at least mentioned that there are no significant El Niño-induced ozone anomalies in the SH extratropical (30S-60S) MLS observations in contrast to the CLaMS simulations and to the NH extratropics.***

Actually, El Niño-induced ozone anomalies in the SH extratropics (30S-60S) are also present in MLS observations with a weaker amplitude than the CLaMS simulations. We have rephrased the sentence. Page 7, line 27-32.

6. ***P.7, L30-31: "In the extratropical UTLS (3070), CLaMS model and MLS observations show a related positive O3 anomaly due to enhanced downwelling." In the LS, MLS observations show only positive ozone anomalies in the NH extratropics (see Fig.3). This is not true for SH extratropics (see general comments). It is true that CLaMS simulations with ERA-I show enhanced downwelling in SH explaining the ozone simulations, but not the observations in the SH.***

By simply plotting the MLS ozone anomalies on a color bar scale 2 time smaller than CLaMS color bar scale in Fig. 3, we see now more clearly the positive ozone anomalies in the SH extratropics. We have rephrased the paragraph. Page 8, line 7-22.

7. ***P.7, L. 35: Could you please explain, why ozone anomalies above 500K are affected by upper boundary conditions and why they are not affected below.***

The upper boundary conndition for CLaMS ozone is set at 500K. Above 500K, the ozone in CLaMS is just the mean climatological values, thus excluding representation of variability. We have rephrased it.

8. ***P.8, L13: Again, only missing tropospheric chemistry and lower boundary conditions are to my opinion not sufficient to explain MLS vs. CLaMS-ERA-I differences in Fig. 2 and Fig 3 (see general comments). Or their impact on the ozone anomalies should be quantified somehow.***

Please see the answer to general comments.

9. ***P.8, L23-25: "The picture of negative AoA anomalies in the tropical lower stratosphere and positive AoA anomalies in the mid and high latitudes (30-60 N and S) agrees well with O3 anomalies from CLaMS simulations and MLS observations (Fig. 3)." Yes, the picture of AoA and ozone anomalies simulated with CLaMS-ERA-I is consistent, but the picture is not consistent for MLS observations of ozone anomalies in SH extratropical (30-60S) LS.***

The picture of negative AoA anomalies in the tropical lower stratosphere and positive AoA anomalies in the mid and high latitudes (30-60 N and S) is more or less consistent with ozone anomalies from CLaMS simulations and MLS observations (Fig. 3), although, the MLS observations of ozone changes show weaker anomalies in SH extratropical (30-60S) lower stratosphere than the reanalyses. We have rephrased it. Page 9, line 10-12

10. ***P.9, L.16-17: ..., while the shallow branch is strengthening in both reanalyses. Assuming 420 to 550K as the shallow branch, it seems that RCTT and AoA (residual and tracer transport) from JRA55 is not really indicative for a strengthening of the shallow branch. This statement depends strongly on the definition of both branches in potential temperature coordinates (see also general comments).***

RCTT and AoA (residual and tracer transport) from JRA-55 indicate a strengthening of the shallow branch but the shallow branch is more confined to the tropics than it is in ERA-I. We have rephrased the comments as: "The pattern of changes in the residual circulation (transit time and stream function) depicts a weakening transition branch during El Niño, while the shallow branch is strengthening in both reanalyses. However, differences occur between the two reanalyses concerning the strength of the shallow branch. The strengthening of the shallow branch in response to El Niño does not extend as far poleward in JRA-55 as it does in ERA-I, reflecting the difference in the strength of the tropical upwelling response in the two reanalyses (Fig. 4(c, d)). Page 10, line 15-20.

11. ***Figure 6: Please use the same range for the left y-axis for both tropical (and for both midlatitude) plots.***

Using the same range for the left y-axis in Fig. 7 will flatten the plots especially for the mid latitude, therefore decreases the clarity of the changes in the young air mass. The percentage of changes in air

mass is given in current Fig. 8. We have used the same y-axis range only for the tropics and added a notice about different y-axis for the mid-latitude plots in the caption.